# EFFICIENT POINT CLOUD MATCHING
# FOR 3D GEOMETRIC SHAPE ASSEMBLY

## ABSTRACT

Learning to assemble geometric shapes into a larger target structure is a fundamental task with various high-level vision applications. In this work, we frame this problem as geometric registration with extremely low overlap. Our goal is to establish accurate correspondences on the mating surface of the shape fragments to predict their relative rigid transformations for assembly. To this end, we introduce Proxy Match Transform (PMT), an approximate high-order feature transform layer that enables reliable correspondences between dense point clouds of shape fragments, while incurring low costs in memory and compute. In our experiments, we demonstrate that Proxy Match Transform surpasses existing state-of-the-art baselines on a popular geometric shape assembly dataset, while exhibiting significantly better efficiency than other high-order feature transform methods.

## 1 INTRODUCTION

Shape assembly is a pivotal task aiming to determine the precise placement and orientation of fractured parts, with the primary objective of constructing a larger target structure. This task holds paramount significance, especially in the context of various applications encompassing robotics (Wang & Hauser, 2019; Zakka et al., 2020; Zeng et al., 2021), manufacturing, computer graphics (Li et al., 2012), and computer-aided design (Chen et al., 2015; Jacobson, 2017). Despite its pivotal role in industrial productivity and the plethora of applications it underpins, shape assembly has received relatively limited attention in the literature. It persists as a formidable challenge due to its inherent complexities. These complexities arise from the necessity to comprehend the intricate geometric structures at play and establish reliable local pairwise relationships between input shape pairs to facilitate accurate assembly.

An overarching challenge in shape assembly lies in the precise identification of mating surfaces and the subsequent verification of local correspondences between the surfaces. Previous methods have addressed these challenges by relying on semantic information linked to object fragments, *e.g.*, part labels, effectively reconstructing complete objects by aligning their semantic components (Narayan et al., 2022; Li et al., 2020b; Huang et al., 2020). However, these methods necessitate additional semantic annotations, which restricts their applicability to semantic shape assembly and specific datasets. To expand the scope of research, Sellán et al. (2022) introduced a large-scale assembly benchmark, called BreakingBad dataset, that simulates physically broken objects resulting from external forces, which facilitates the study of *geometric* shape assembly with diverse objects.

The task of geometric shape assembly can be framed within a broader context of point cloud registration. Recent methods for the task typically utilize a high-order feature transform, *e.g.*, high-order convolution or attention, to establish reliable correspondences from noisy input (Choy et al., 2020; Yu et al., 2021; Huang et al., 2021; Qin et al., 2022). The high-order feature transform considers structural patterns of correlations in a high-dimensional space and has been particularly effective in matching and registration (Rocco et al., 2018; Min & Cho, 2021; Kim et al., 2022). However, a significant challenge arises from its quadratic complexity concerning input resolution, restricting its use only to coarse-grained matching with a limited number of points. In the context of geometric shape assembly, this complexity issue greatly hinders the performance since the object fractures are typically complex and intricate, demanding meticulous registration with dense point clouds.

In this paper, we address this issue by introducing a new form of low-complexity high-order feature transform layer, dubbed *Proxy Match Transform (PMT)*, for use in geometric shape assembly

(Fig. 1). We theoretically prove that the proposed PMT layer can effectively approximate the conventional high-order convolution layers (Rocco et al., 2018; Choy et al., 2020; Min & Cho, 2021) under some conditions. To demonstrate the effect, we incorporate the PMT layer into a coarse-to-fine matching network, where PMT is used in both coarse-level and fine-level matching steps to establish reliable correspondences between mating surfaces and enable accurate assembly. We compare our result with recent state-of-the-art approaches and thoroughly analyze its performance on the geometric shape assembly benchmark, *e.g.*, BreakingBad dataset. The experiments demonstrate that our method outperforms existing approaches while being computationally efficient.

Our main contributions can be summarized as follows:

- We introduce Proxy Match Transform, a low-complexity high-order feature transform layer that effectively refines the matching of the feature pair.

- Our theoretical analysis showcases how Proxy Match Transform approximates high-order convolution while incurring only sub-quadratic complexity.

- The performance improvements in geometric shape assembly over the state-of-the-art baselines demonstrate the efficacy of the proposed approach.

## 2 RELATED WORK

**3D shape generation & assembly.** Previous studies explored the generative models that represent objects through the combination of basic 3D primitives. One approach involves training specialized models for individual object classes, enabling them to assemble objects from volumetric primitives, such as cuboids (Tulsiani et al., 2017). Another approach involves training a single model capable of generating cuboid primitives for all classes (Khan et al., 2019). Variational autoencoders (VAE) (Kingma & Welling, 2014) have also been used to model objects as combinations of cuboids (Jones et al., 2020). These methods provide robust abstractions, distilling local geometric details and revealing object correspondences.

Another related area of research deals with part assembly, aiming to construct complete objects from sets of parts. Li et al. (2020b) proposed to predict translations and rotations for part point clouds to assemble a target object from an image. Narayan et al. (2022); Huang et al. (2020) frame part assembly as graph learning, using iterative message passing to assemble parts into complete objects. These methods leverage the PartNet (Mo et al., 2019) dataset to ensure that assembled parts semantically correspond to the target object. While shape is crucial in part assembly, part semantics can also guide the process directly, bypassing geometric cues.

Our work addresses the problem of learning to fit together pieces with no particular semantics and without a provided target. The most relevant method to ours is Chen et al. (2022), which aims to solve 3D shape assembly while coupling the task with implicit shape reconstruction. In contrast, we aim to formulate this task as an extremely low-overlap point cloud registration, eschewing the need for shape reconstruction and rather focusing on finding reliable correspondences between the shapes to fit together along their mating surface.

**High-order feature transform.** The idea of high-order feature transform has been widely applied in areas handling visual correspondence to identify the consensus among correspondences in a high-dimensional space. Rocco et al. (2018) coined the idea of learning-based neighborhood consensus, such that a certain match will support its neighboring ambiguous matches between 2D images.

To alleviate the high computation complexity of high-order feature transform, subsequent studies propose to squeeze the high-dimensional correlation (Min et al., 2021), sparsify the correlation map using top-k scores to reduce computation (Rocco et al., 2020). More recently, Cho et al. (2021) and Kim et al. (2022) proposed to integrate the self-attention mechanism to leverage the global consensus between the high-dimensional features, which have shown to be effective albeit at an increased cost.

A similar trend is evident in the area of 3D registration; Choy et al. (2020) aims to filter outlier correspondences using a 6D sparse convolutional layer (Choy et al., 2019), while Huang et al. (2021) and Yu et al. (2021) aims to transform 3D features by leveraging self-attention and cross-attention, both within and across point cloud features. Qin et al. (2022) proposed to embed transformation-invariant information into the positional embedding of the transformer layers, achieving robust su-

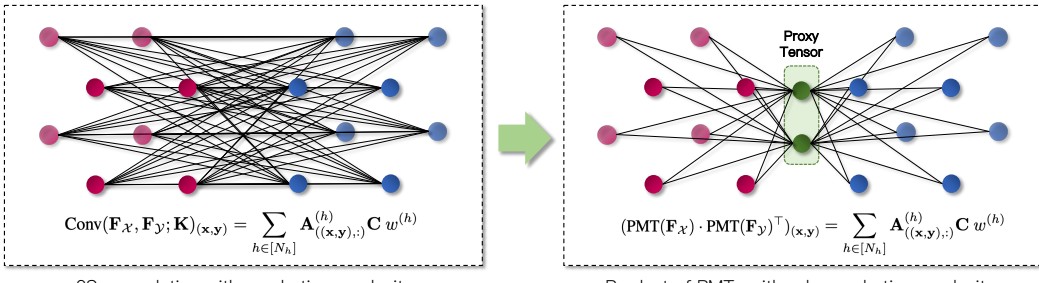

$$\text{Conv}(\mathbf{F}_{\mathcal{X}}, \mathbf{F}_{\mathcal{Y}}; \mathbf{K})_{(\mathbf{x},\mathbf{y})} = \sum_{h \in [N_h]} \mathbf{A}^{(h)}_{((\mathbf{x},\mathbf{y}),:)} \mathbf{C}\, w^{(h)}$$

2S–convolution with quadratic complexity

$$(\text{PMT}(\mathbf{F}_{\mathcal{X}}) \cdot \text{PMT}(\mathbf{F}_{\mathcal{Y}})^{\top})_{(\mathbf{x},\mathbf{y})} = \sum_{h \in [N_h]} \mathbf{A}^{(h)}_{((\mathbf{x},\mathbf{y}),:)} \mathbf{C}\, w^{(h)}$$

Product of PMTs with sub–quadratic complexity

Figure 1: Given a pair of $S$-dimensional features, dot product of our proposed feature transform layers, *i.e.*, Proxy Match Transform (PMT), can express the $2S$-convolution with sub-quadratic complexity, while existing high-order convolution (Rocco et al., 2018; Choy et al., 2020; Min & Cho, 2021) needs quadratic complexity.

perpoint matching accuracy. While these methods excel in low-overlap scenarios, their adoption for fine-grained matching is hindered by inherent high computational costs.

In this work, we propose Proxy Match Transform (PMT), which approximates existing high-order feature transforms while incurring significantly lower computation complexity. We integrate PMT in a coarse-to-fine manner, where on coarse-level PMT identifies accurate and reliable correspondences on the mating surface of input shape fragments, then refines on fine-level for more precise assembly.

# 3 PROPOSED APPROACH

In the task of geometric shape assembly, analyzing geometric compatibility between fractured shapes is of utmost importance; the geometric properties of the *mating surfaces* should exhibit consistency, where vertices, edges, and surfaces seamlessly fit together to form a coherent structure. To achieve reliable localization of mating surfaces between shapes, a model needs to analyze the compatibility of all possible feature correspondences and accurately identify spatially consistent matches. In the field of matching and registration (Rocco et al., 2018; Choy et al., 2020; Min & Cho, 2021) and its applications (Min et al., 2021), a trending approach for assessing match reliability is the utilization of **high-order convolution**. This technique effectively assesses patterns within neighborhood matches in a differentiable manner. Building upon these principles, we will now explore the theoretical formulation of high-order convolution, with a specific emphasis on its application for enhancing pairwise feature correlation.

In the section, we first start with the preliminary where we revisit the concept of high-order convolution and introduce the theorem that multi-head self-attention (MHSA) can express convolution; this motivates us to use an attention-based formulation to approximate the high-order convolution. We then present the Proxy Match Transform (PMT) and show how PMT can effectively express high-order convolution with only sub-quadratic complexity (Sec. 3.1). Finally, we explain two constraints of PMT for its approximation (Sec. 3.2). The necessary proofs and discussions supporting our approach are further detailed in the Appendix, and each will be referenced throughout the relevant sections.

**Preliminary.** High-order convolution (Rocco et al., 2018; Choy et al., 2020; Min & Cho, 2021) generalizes the standard convolution by taking as input more functions, feature maps, or sets. In the context of our problem, we consider two point clouds $\mathcal{X} = \{\mathbf{x}_i \in \mathbb{R}^3 | i = 1, ..., N\}$ and $\mathcal{Y} = \{\mathbf{y}_i \in \mathbb{R}^3 | i = 1, ..., M\}$, and focus on the 2nd-order convolution with two sets of features $\mathcal{F}_{\mathcal{X}}$ and $\mathcal{F}_{\mathcal{Y}}$, associated with the two point clouds, respectively. For ease of notation, we represent these features in matrix form, *i.e.*, $\mathbf{F}_{\mathcal{X}} \in \mathbb{R}^{|\mathcal{X}| \times D_{\text{emb}}}$, where $D_{\text{emb}}$ is the feature embedding dimension, and indexes each feature of the matrix using its associated point $\mathbf{x} \in \mathcal{X}$ such that $\mathbf{F}_{\mathbf{x}} \in \mathbb{R}^{D_{\text{emb}}}$, and same goes for $\mathcal{F}_{\mathcal{Y}}$. We also express the feature correlation of two points from each point cloud, $\mathbf{x}$ and $\mathbf{y}$,

as $\mathbf{C}_{(\mathbf{x},\mathbf{y})} := \mathbf{F}_\mathbf{x}\mathbf{F}_\mathbf{y}^\top$. The 2nd-order convolution on $(\mathbf{F}_\mathcal{X}, \mathbf{F}_\mathcal{Y})$ with kernel $K$ is then defined as

$$\text{Conv}(\mathbf{F}_\mathcal{X}, \mathbf{F}_\mathcal{Y}; \mathbf{K})_{(\mathbf{x},\mathbf{y})} := \sum_{(\mathbf{n},\mathbf{m})\in\mathcal{N}(\mathbf{x})\times\mathcal{N}(\mathbf{y})} \mathbf{C}_{(\mathbf{n},\mathbf{m})}K([\mathbf{n}-\mathbf{x}, \mathbf{m}-\mathbf{y}]), \tag{1}$$

where $\mathcal{N}(\cdot)$ represents a set of neighbor points and $K : \mathbb{R}^6 \to \mathbb{R}$ is a convolutional kernel, represented as a mapping function that takes displacement vectors onto learnable weight scalar.

On the other hand, Cordonnier et al. (2020) show the relation of self-attention to convolution:

**Theorem (Cordonnier et al., 2020).** *A multi-head self-attention layer with $N_h$ heads of dimension $D_h$, output dimension $D_{out}$ and a relative positional encoding of dimension $D_p \geq 3$ can express any convolutional layer of kernel size $\sqrt{N_h} \times \sqrt{N_h}$ and $\min(D_h, D_{out})$ output channels.*

Motivated by this, we attempt to express the high-order convolution using an attention-based form.

As illustrated in Fig. 1, the 2nd-order convolution (Eq. 1) disambiguates spatially consistent correspondences and updates their correlation values by analyzing correlation patterns around each point pair $(\mathbf{x}, \mathbf{y}) \in \mathcal{X} \times \mathcal{Y}$. Despite its good empirical performance in literature (Rocco et al., 2018; Choy et al., 2020; Min & Cho, 2021; Min et al., 2021), its critical limitation lies in the quadratic complexity of correlation computation, *i.e.*, $\mathcal{O}(|\mathcal{X}| \cdot |\mathcal{Y}|)$, with respect to input resolution, imposing significant computational burdens during both the training and inference phases. This restricts its practical applications for large spatial resolutions; Notably, the geometric shape assembly task demands sophisticated matching techniques, especially at high resolutions, to ensure precise assembly.

## 3.1 PROXY MATCH TRANSFORM: EFFICIENT HIGH-ORDER FEATURE TRANSFORM WITH SUB-QUADRATIC COMPLEXITY

To overcome this limitation, we introduce an efficient correlation refinement layer called *Proxy Match Transform (PMT)*, which can effectively express high-order convolution with sub-quadratic complexity. Given a pair of input features $(\mathbf{F}_\mathcal{X}, \mathbf{F}_\mathcal{Y})$, two of Proxy Match Transforms with $N_h$ heads[1] are defined for input feature, $\mathbf{F}_\mathcal{X}$ and $\mathbf{F}_\mathcal{Y}$, respectively, as follows:

$$\text{PMT}(\mathbf{F}_\mathcal{X}) := \sum_{h\in[N_h]} \mathbf{A}_\mathcal{X}^{(h)}\mathbf{F}_\mathcal{X}\mathbf{P}^{(h)\top}w_\mathcal{X}^{(h)}, \tag{2}$$

$$\text{PMT}(\mathbf{F}_\mathcal{Y}) := \sum_{h\in[N_h]} \mathbf{A}_\mathcal{Y}^{(h)}\mathbf{F}_\mathcal{Y}\mathbf{P}^{(h)\top}w_\mathcal{Y}^{(h)}, \tag{3}$$

where $w_\mathcal{X}^{(h)} \in \mathbb{R}$ is a learnable weight scalar, $\mathbf{A}_\mathcal{X}^{(h)} \in \mathbb{R}^{|\mathcal{X}|\times|\mathcal{X}|}$ is local attention matrix [2]; the same applies for $\mathcal{Y}$. And $\mathbf{P}^{(h)} \in \mathbb{R}^{D_{\text{proxy}}\times D_{\text{emb}}}$ is **proxy tensor** that satisfies:

$$\mathbf{P}^{(h)\top}\mathbf{P}^{(h)} = \mathbf{I}_{D_{\text{emb}}}, \ \forall_{h\in[N_h]}, \tag{4}$$

where $D_{\text{proxy}}$ refers to the spatial resolution of the proxy tensor: $D_{\text{proxy}} \ll |\mathcal{X}|$ and $D_{\text{proxy}} \ll |\mathcal{Y}|$. We refer the readers to Appendix A.4 for the details of attention calculation.

At each head, the layer initially constructs a correlation between the input feature $\mathbf{F}_\mathcal{X}$ and the proxy tensor $\mathbf{P}^{(h)}$: $\mathbf{C}_\mathcal{X}^{(h)} := \mathbf{F}_\mathcal{X}\mathbf{P}^{(h)\top}$ in much smaller size of $|\mathcal{X}| \times D_{\text{proxy}}$, compared to the pairwise feature correlation $\mathbf{C} = \mathbf{F}_\mathcal{X}\mathbf{F}_\mathcal{Y}^\top \in \mathbb{R}^{|\mathcal{X}|\times|\mathcal{Y}|}$ defined in Eq. 1. After applying learnable weight $w_\mathcal{X}^{(h)}$, the output at position $(\mathbf{n}, \mathbf{m}) \in |\mathcal{X}| \times D_{\text{proxy}}$ is computed through a weighted-sum of its neighborhood matches lying on the spatial dimension of feature map $\mathbf{F}_\mathcal{X}$, *e.g.*, $\{(\mathbf{n}', \mathbf{m})\}_{\mathbf{n}'\in\mathcal{N}(\mathbf{n})}$ where $|\mathcal{N}(\mathbf{n})| = \epsilon \ll |\mathcal{X}|$. To formally put, the Proxy Match Transform output at head $h$ given

---

[1]Similar to multi-head self-attention (Vaswani et al., 2017), each head performs distinct attentions and feature transform, allowing the layer to attend and transform different aspects of the input simultaneously.

[2]As attention matrix $\mathbf{A}_\mathcal{X}^{(h)}$ is local, *i.e.*, sparse, in actual implementation, we reduce its size to $|\mathcal{X}| \times \epsilon$ instead of $|\mathcal{X}| \times |\mathcal{X}|$ where $\epsilon \in \mathbb{N}^+$ is the number of neighbors: $|\mathcal{X}| \gg \epsilon$. For ease of presentation, we demonstrate our method using square attention matrix $\mathbf{A}_\mathcal{X}^{(h)} \in \mathbb{R}^{|\mathcal{X}|\times|\mathcal{X}|}$.

input $\mathbf{F}_{\mathcal{X}}$ at position $(\mathbf{n}, \mathbf{m})$ is defined as

$$
\begin{aligned}
\text{PMT}(\mathbf{F}_{\mathcal{X}})^{(h)}{}_{(\mathbf{n},\mathbf{m})} &= \mathbf{A}_{\mathcal{X}}^{(h)}{}_{(\mathbf{n},:)} \mathbf{F}_{\mathcal{X}} \mathbf{P}^{(h)\top}{}_{(:,\mathbf{m})} w_{\mathcal{X}}^{(h)} = \mathbf{A}_{\mathcal{X}}^{(h)}{}_{(\mathbf{n},:)} \mathbf{C}_{\mathcal{X}}^{(h)}{}_{(:,\mathbf{m})} w_{\mathcal{X}}^{(h)} \\
&= \sum_{\mathbf{n}' \in \mathcal{N}(\mathbf{n})} \mathbf{A}_{\mathcal{X}}^{(h)}{}_{(\mathbf{n},\mathbf{n}')} \mathbf{C}_{\mathcal{X}}^{(h)}{}_{(\mathbf{n}',\mathbf{m})} w_{\mathcal{X}}^{(h)},
\end{aligned}
\tag{5}
$$

where $\text{PMT}(\mathbf{F}_{\mathcal{Y}})^{(h)}$ is similarly defined but with a different set of parameters of $\mathbf{A}_{\mathcal{Y}}^{(h)}$ and $w_{\mathcal{Y}}^{(h)}$.

It is important to note that Proxy Match Transform layer performs two ***independent*** transform for ***feature matching***, one for $\mathbf{F}_{\mathcal{X}}$ and the other for $\mathbf{F}_{\mathcal{Y}}$. Despite the independence, matching between the feature pair is effectively facilitated by a shared proxy tensor $\mathbf{P}$. This proxy tensor allows for the exchange of information between the features, eliminating the need to construct and convolve memory-intensive pairwise feature correlations, which often contain sparse and limited informative match scores. Additionally, it is worth noting that different sets of parameters are used for $\mathbf{F}_{\mathcal{X}}$ and $\mathbf{F}_{\mathcal{Y}}$ which enhances the flexibility and adaptability for matching. In Sec. 4.2, we empirically prove the efficacy of the use of proxy tensor and different parameter sets in point cloud matching.

## 3.2 Constraints for Proxy Match Transform

In order for the Proxy Match Transforms to express the high-order convolution, we assume two constraints, (i) orthonormality constraint: $\mathbf{P}^{(i)\top}\mathbf{P}^{(j)} = \mathbf{I}_{D_{\text{emb}}}$ if $i = j$, and (ii) zero constraint: $\mathbf{P}^{(i)\top}\mathbf{P}^{(j)} = \mathbf{0} \in \mathbb{R}^{D_{\text{emb}} \times D_{\text{emb}}}$ otherwise for all $i, j \in [N_h]$. Given these constraints on proxy tensors, the dot product between PMT outputs effectively approximates the high-order convolution:

$$
(\text{PMT}(\mathbf{F}_{\mathcal{X}}) \cdot \text{PMT}(\mathbf{F}_{\mathcal{Y}})^{\top})_{(\mathbf{x},\mathbf{y})} \approx \text{Conv}(\mathbf{F}_{\mathcal{X}}, \mathbf{F}_{\mathcal{Y}}; \mathbf{K})_{(\mathbf{x},\mathbf{y})}.
\tag{6}
$$

We refer the readers to the Appendix A.1, for the complete proof. For the proxy tensors to satisfy the conditions, we design two auxiliary training objectives on proxy tensors, orthonormal loss $\mathcal{L}_{orth}$ and zero loss $\mathcal{L}_{zero}$, as follows:

$$
\mathcal{L}_{orth} = \sum_{(i,j) \in [N_h]^2} \delta(i,j)(\mathbf{P}^{(i)\top}\mathbf{P}^{(j)} - \mathbf{I}_{D_{\text{emb}}}),
\tag{7}
$$

$$
\mathcal{L}_{zero} = \sum_{(i,j) \in [N_h]^2} (1 - \delta(i,j))\mathbf{P}^{(i)\top}\mathbf{P}^{(j)},
\tag{8}
$$

where $\delta(i,j)$ provides 1 if $i = j$ and 0 otherwise.

## 3.3 Overall Architecture

Fig. 2 illustrates the overall architecture of our method. Building on recent developments in the field, *e.g.*, GeoTransformer (Qin et al., 2022), it introduces two key enhancements to optimize its performance: (i) replacing the coarse-level matcher with Proxy Match Transform layers, and (ii) incorporating Proxy Match Transform layers at the fine-level stages after each upsampling in KPConv-FPN backbone (Thomas et al., 2019) as seen in Fig. 2.

**Feature extractor.** Given a pair of mesh fragments for pairwise assembly, we first sample a total of 5,000 points from an object $O$. The sizes of the point sets $\mathcal{X}$ and $\mathcal{Y}$ are then determined based on the proportional volumes each mesh fragment occupies within the entire object $O$. These point sets are then fed into the KPConv-FPN backbone, which downsamples the input twice, yielding coarse-level feature pair $\hat{\mathbf{F}}_{\mathcal{X}} \in \mathbb{R}^{|\hat{\mathcal{X}}| \times D_{\text{c}}}$ and $\hat{\mathbf{F}}_{\mathcal{Y}} \in \mathbb{R}^{|\hat{\mathcal{Y}}| \times D_{\text{c}}}$, and subsequently upsamples twice to obtain fine-level feature pair $\tilde{\mathbf{F}}_{\mathcal{X}} \in \mathbb{R}^{|\tilde{\mathcal{X}}| \times D_{\text{f}}}$ and $\tilde{\mathbf{F}}_{\mathcal{Y}} \in \mathbb{R}^{|\tilde{\mathcal{Y}}| \times D_{\text{f}}}$.

Note that in the case of fine-level features $\tilde{\mathbf{F}}_{\mathcal{X}}$ and $\tilde{\mathbf{F}}_{\mathcal{Y}}$, refinement is carried out at the upsampling stage through the ***fine matcher***, *i.e.*, two PMT blocks, each consisting of multiple PMT layers followed by a sequence of Group Normalization and LeakyReLU, and repeats for $N_t$ times. It is important to note that the performance of the fine matcher is a critical aspect of our approach, and we will present empirical results to demonstrate its effectiveness in Sec. 4.2.

**Coarse-level Matching.** As seen in Fig. 2, at both coarse and fine levels, distinct Proxy Match Transform layers are applied to establish correspondences between fragments. Similarly to the fine

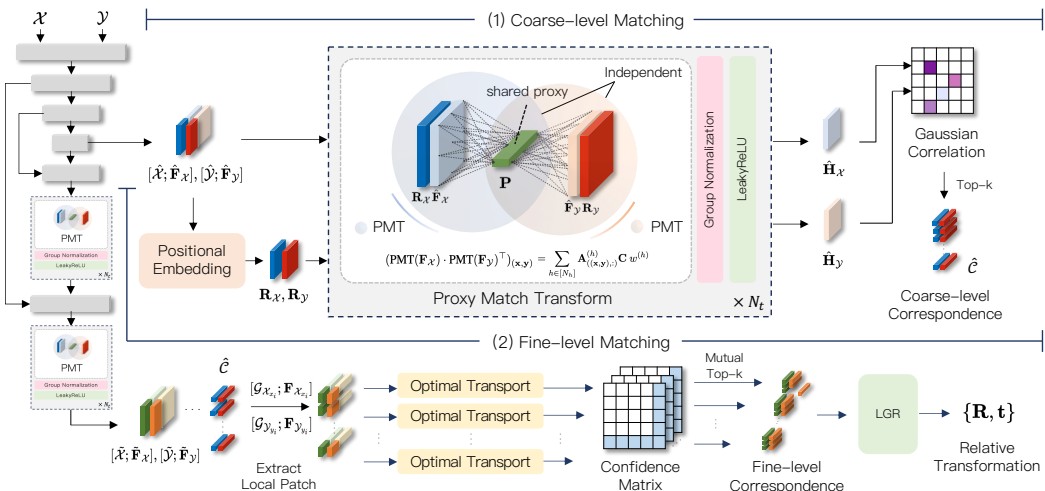

Figure 2: Overall pipeline of the proposed Proxy Match Transform for pairwise shape assembly. The proposed architecture largely consists of two modules: (1) coarse-level matching and (2) fine-level matching. Each module uses coarse-level features and fine-level features acquired from the KPConv-FPN backbone as their input. Details are described in Sec. 3.3

matcher, the *coarse matcher* also utilizes a PMT block with an identical configuration. The inputs for the PMT block in the coarse matcher are the coarse-level features $\hat{\mathbf{F}}_{\mathcal{X}}$ and $\hat{\mathbf{F}}_{\mathcal{Y}}$, together with their respective positional embeddings $\mathbf{R}_{\mathcal{X}}$ and $\mathbf{R}_{\mathcal{Y}}$. After the PMT block as the coarse matcher, the output feature pair, denoted as $\hat{\mathbf{H}}_{\mathcal{X}}$ and $\hat{\mathbf{H}}_{\mathcal{Y}}$, is utilized to compute the pairwise similarity. We then select the top-k matches as coarse-level correspondences $\hat{\mathcal{C}}$.

**Fine-level Matching.** Once we have reliable matches at the coarse level, we apply the point-to-node grouping method (Yu et al., 2021). This technique allows us to expand the coarse-level correspondences to fine-level correspondences and to extract local patches based on these coarse-level correspondences, denoted as $\hat{\mathcal{C}}$. Subsequently, these local patches are fed into an optimal transport layer (Sarlin et al., 2020), which facilitates the extraction of confidence matrices. Using these matrices, we obtain fine-level correspondences by selecting the mutual top-k entries (Qin et al., 2022).

Following this, Local-to-Global Registration (LGR) (Qin et al., 2022) takes these fine-level correspondences into account, to select and refine the final relative transformation, represented as $\{\mathbf{R}|\mathbf{t}\}$.

**Training objectives.** Following Qin et al. (2022), we adopt training objectives to establish coarse- and fine-level correspondences, which are, respectively, the circle loss objective $\mathcal{L}_{oc}$ and point matching loss objective $\mathcal{L}_p$. We refer the readers to Qin et al. (2022) for details on the framework.

Our final training objective is described as follows:

$$\mathcal{L} = \mathcal{L}_{oc} + \mathcal{L}_p + \lambda_{orth}\mathcal{L}_{orth} + \lambda_{zero}\mathcal{L}_{zero}, \qquad (9)$$

where $\lambda_{orth}$ and $\lambda_{zero}$ are hyperparameters that manage the contribution of each respective loss. In our case, we have set both of these hyperparameters to 1.

## 4 EXPERIMENTS

In this section, we evaluate our proposed method, compare it with the recent state of the arts, and provide in-depth analyses of the results with ablation study in Sec. 4.2. We first show the results on pairwise shape assembly in Sec. 4.1, and with considering pairwise assemble as a partial stage of multi-part assembly, then show the extended experimental results in Sec. 4.3.

### 4.1 PAIRWISE SHAPE ASSEMBLY

**Dataset.** We use the BreakingBad dataset (Sellán et al., 2022) to train and evaluate Proxy Match Transform on the task of pairwise shape assembly. The BreakingBad dataset is a large-scale dataset

Table 1: Pairwise shape assembly results on BreakingBad dataset.

| Method | CRD ↓ ($10^{-2}$) | CD ↓ ($10^{-3}$) | RMSE (R) ↓ (°) | RMSE (T) ↓ ($10^{-2}$) | CRD ↓ ($10^{-2}$) | CD ↓ ($10^{-3}$) | RMSE (R) ↓ (°) | RMSE (T) ↓ ($10^{-2}$) |
|---|---|---|---|---|---|---|---|---|
| | everyday | | | | artifact | | | |
| ICP | 8.57 | 6.52 | 83.73 | 27.41 | 7.48 | 5.01 | 83.41 | 25.11 |
| Sparse ICP | 8.55 | 7.08 | 83.54 | 27.26 | 8.49 | 6.37 | 83.73 | 25.06 |
| RANSAC | 8.25 | 5.39 | 88.62 | 26.34 | 7.44 | 4.49 | 83.43 | 26.82 |
| FGR | 8.05 | 6.44 | 84.65 | 26.64 | 7.23 | 4.91 | 83.39 | 24.81 |
| Global | 25.10 | 13.80 | 86.40 | 28.80 | 24.70 | 12.39 | 87.59 | 26.69 |
| LSTM | 27.90 | 15.20 | 85.20 | 30.70 | 24.07 | 11.12 | 83.86 | 25.84 |
| DGL | 24.20 | 12.10 | 85.50 | 28.10 | 24.62 | 10.68 | 86.03 | 26.36 |
| NSM | 22.23 | 14.54 | 80.83 | 24.85 | 23.18 | 9.97 | 83.97 | 11.37 |
| GeoTransformer | **1.66** | 1.23 | 34.40 | 8.82 | 2.65 | 2.22 | 46.63 | 11.37 |
| **PMT (Ours)** | 1.67 | **1.05** | **33.40** | **8.16** | **2.46** | **2.08** | **40.08** | **10.39** |

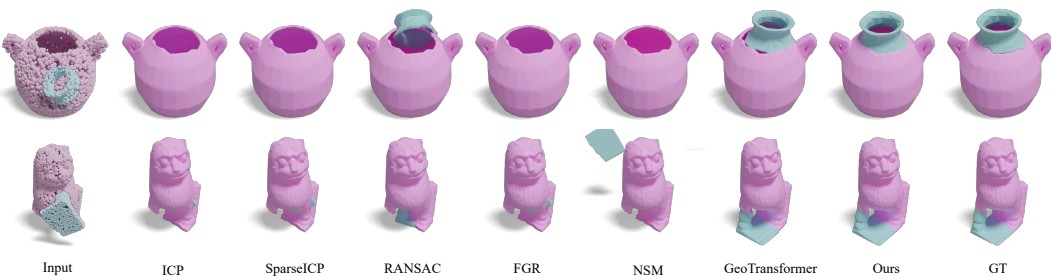

Figure 3: Qualitative results of pairwise shape assembly on BreakingBad dataset.

of fractured objects for the task of geometric shape assembly, which consists of over 1 million fractured objects simulated from 10K meshes of PartNet (Mo et al., 2019) and Thingi10k (Zhou & Jacobson, 2016). Our focus, akin to the work of Chen et al. (2022), is centered on the problem of mating a pair of fragments. Please note that we exclusively select a subset that contains two-fragment objects from the BreakingBad dataset for training and evaluating pairwise assembly, while we utilize all samples for training and evaluating multi-part assembly.

**Baselines.** To assess the performance of our method, we conduct a comprehensive comparison with three main groups of baseline methods. First, for non-learning-based point cloud registration methods, we used the ICP (Besl & McKay, 1992), and SparseICP (Bouaziz et al., 2013) for local registration methods, and RANSAC (Fischler & Bolles, 1981) and FGR (Zhou et al., 2016) for global registration. Next, we also include four learning-based shape assembly methods, namely Global (Li et al., 2020a), LSTM (Zhang et al., 2019), DGL (Huang et al., 2020), and NSM (Chen et al., 2022). The first three methods are semantic shape assembly techniques that served as baselines in a previous study by Sellán et al. (2022). NSM (Chen et al., 2022) is, as of our submission, the only learning-based baseline available for the geometric shape assembly task. Lastly, we compare with the recent learning-based point cloud registration method, GeoTransformer (Qin et al., 2022).

**Evaluation metrics.** Following the evaluation protocol of Sellán et al. (2022), we measure the root mean square error (RMSE) between the ground-truth and predicted rotation and translation parameters, and the Chamfer distance (CD) between the assembly results and ground-truth. In addition, we introduce and report a new metric, called CoRrespondence Distance (CRD), which is simply defined as the Frobenius norm between two point clouds. Unlike Chamfer distance, it offers a more comprehensive measure of similarity, capturing both proximity and structural alignment. For a detailed definition of each metric, please refer to the Appendix A.5.

**Implementation details.** We implement our Proxy Match Transform using Pytorch (Paszke et al., 2019). Experiments were conducted on a machine with Intel(R) Xeon(R) Gold 6342 CPU @ 2.80GHz and NVIDIA GeForce RTX 3090 GPU. For all experiments, except the ones include Geo-Transformer, we use ADAM (Kingma & Ba, 2015) optimizer with an initial learning rate of $1 \times 10^{-3}$ without learning rate decay for 300 epochs on a single GPU. For GeoTransformer, we use the iden-

Table 2: **Ablation study on the proxy sharing.** By sharing proxy tensor in each Proxy Match Transform layer, two independent feature transforms share information, yielding the highest score.

| Ref. | proxy | shared proxy | CRD↓ $(10^{-2})$ | CD↓ $(10^{-3})$ | RMSE (R)↓ (°) | RMSE (T)↓ $(10^{-2})$ |
|------|-------|--------------|------|------|------|------|
| (a) | ✗ | ✗ | 4.27 | 3.77 | 63.96 | 17.28 |
| (b) | ✓ | ✗ | 2.06 | 1.36 | 38.02 | 8.80 |
| **Ours** | ✓ | ✓ | **1.67** | **1.05** | **33.40** | **8.16** |

Table 3: **Ablation study on the contribution of $\mathcal{L}_{orth}$ and $\mathcal{L}_{zero}$.** Both losses effectively constrains Proxy Match Transform in approximating the high-order convolution layers, yielding the highest score.

| Ref. | orth loss | zero loss | CRD↓ $(10^{-2})$ | CD↓ $(10^{-3})$ | RMSE (R)↓ (°) | RMSE (T)↓ $(10^{-2})$ |
|------|-----------|-----------|------|------|------|------|
| (a) | ✗ | ✗ | 1.81 | 1.28 | 35.27 | 8.23 |
| (b) | ✓ | ✗ | 1.85 | 1.34 | 34.88 | 8.19 |
| (c) | ✗ | ✓ | 1.77 | 1.22 | 35.23 | 8.23 |
| **Ours** | ✓ | ✓ | **1.67** | **1.05** | **33.40** | **8.16** |

Table 4: **Ablation study on the choice of a fine matcher.** (a) Proxy Match Transform layer at fine-level yields the best assembly accuracy while incurring low-compute complexity than baselines. (b) Proxy Match Transform layer consistently delivers improvements, irrespective of the choice of coarse matcher.

(a) Proxy Match Transform as coarse matcher + various fine matchers

| Ref. | Coarse Matcher | Fine Matcher | CRD↓ $(10^{-2})$ | CD↓ $(10^{-3})$ | RMSE(R)↓ (°) | RMSE(T)↓ $(10^{-2})$ |
|------|----------------|--------------|------|------|------|------|
| (a) | | None | 1.92 | 1.48 | 34.41 | 8.68 |
| (b) | | Linear | 2.11 | 1.73 | 37.36 | 9.42 |
| (c) | PMT | MLP | 1.95 | 1.56 | 35.29 | 9.06 |
| (d) | | HDC | Out of memory error | | | |
| (e) | | GeoTr | Out of memory error | | | |
| **Ours** | | PMT | **1.67** | **1.05** | **33.40** | **8.16** |

(b) Various coarse matchers + Proxy Match Transform as fine matcher

| Ref. | Coarse Matcher | Fine Matcher | CRD↓ $(10^{-2})$ | CD↓ $(10^{-3})$ | RMSE(R)↓ (°) | RMSE(T)↓ $(10^{-2})$ |
|------|----------------|--------------|------|------|------|------|
| (a) | None | None | 2.77 | 2.42 | 42.18 | 11.14 |
| (b) | | PMT | **2.10** | **1.68** | **36.41** | **8.32** |
| (c) | Linear | None | 2.22 | 1.90 | 36.50 | 9.18 |
| (d) | | PMT | **1.90** | **1.47** | **33.69** | **8.03** |
| (e) | MLP | None | 2.24 | 1.74 | 36.40 | 9.75 |
| (f) | | PMT | **1.75** | **1.21** | **32.32** | **8.04** |
| (g) | HDC | None | 2.03 | 1.69 | 35.96 | 9.24 |
| (h) | | PMT | **1.64** | **1.17** | **32.42** | **7.86** |
| (i) | GeoTr | None | 1.66 | 1.23 | 34.40 | 8.82 |
| (j) | | PMT | **1.49** | **1.00** | **32.49** | **8.10** |
| (k) | PMT | None | 1.92 | 1.48 | 34.41 | 8.68 |
| **Ours** | | PMT | **1.67** | **1.05** | **33.40** | **8.16** |

tical settings but only reduce the learning rate to $1 \times 10^{-4}$ to prevent model divergence. To ensure uniform point density among fractures, we uniformly sample 5,000 points on the surface of holistic objects and allocate the number of sample points for each fracture proportional to the volume of each fracture. Detailed configurations for all experiments can be found in Appendix A.4.

**Results.** We evaluate our method and compare it against baseline methods on the `everyday` and `artifact` subsets of the BreakingBad dataset. Tab. 1 displays the results, showing that our method consistently outperforms all baseline methods on both subsets. In Fig. 3, we provide qualitative results for all methods. Note that although all methods receive pairs of point clouds, as in the leftmost column of the figure, we present the results using mesh representation for better visualization.

## 4.2 ABLATION STUDY

To evaluate our design choices, we conducted a series of ablation studies. Evaluations for ablation studies are conducted on the `everyday` subset of the BreakingBad dataset. First, we aimed to highlight the impact of the shared proxy tensor in facilitating inter-fragment information exchange. To this end, we conducted an ablation study by removing the shared proxy. The results, as shown in Tab. 2, clearly indicate that eliminating proxy sharing leads to a significant decline in assembly performance, underscoring the efficacy of the shared proxy for facilitating information exchange.

Next, we delve into the impact of $\mathcal{L}_{orth}$ and $\mathcal{L}_{zero}$, which serve as the sufficient conditions that constrain the Proxy Match Transform layer to represent the high-dimensional convolutional layers, as detailed in Sec. 3.2. The results are presented in Tab. 3. As evident from the table, the best performance is achieved when both losses are incorporated. This underscores the significance of these constraining conditions for PMT, as they are instrumental in enabling PMT to effectively approximate the high-dimensional convolutional layers.

Finally, we conduct an ablation study to assess the significance of the PMT layer when employed as a fine matcher. To accomplish this, we conducted two distinct sets of experiments. Initially, we examined the assembly performance by altering the fine-level while fixing the coarse matcher as PMT (Tab. 4 (a)). Subsequently, we incorporated various coarse matchers and then evaluated the

Table 5: Multi-part assembly results on the Breaking Bad dataset.

| Method | CRD↓ (10⁻²) | CD↓ (10⁻³) | RMSE (R)↓ (°) | RMSE (T)↓ (10⁻²) | PA↑ (%) | CRD↓ (10⁻²) | CD↓ (10⁻³) | RMSE (R)↓ (°) | RMSE (T)↓ (10⁻²) | PA↑ (%) |
|---|---|---|---|---|---|---|---|---|---|---|
| | everyday | | | | | artifact | | | | |
| Global | 38.74 | 19.25 | 81.23 | 15.96 | 17.63 | 39.84 | 20.66 | 84.18 | 16.26 | 7.23 |
| LSTM | 40.71 | 23.62 | 85.64 | 16.41 | 11.21 | 41.40 | 27.48 | 85.23 | 16.86 | 2.30 |
| DGL | 38.21 | 18.14 | 81.81 | 15.35 | 20.08 | 40.00 | 20.62 | 86.40 | 16.05 | 6.68 |
| **PMT w/ PGO (Ours)** | **13.81** | **12.79** | **48.22** | **15.27** | **47.82** | **13.91** | **12.29** | **50.76** | **15.52** | **45.14** |

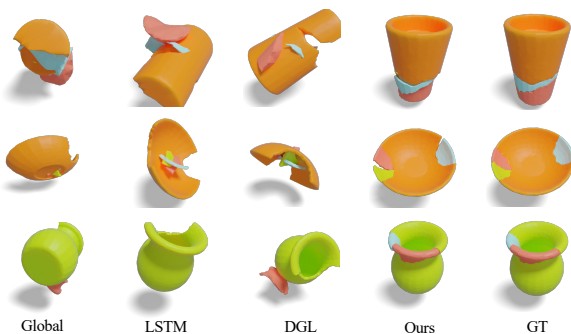

Global          LSTM          DGL          Ours          GT

Figure 4: Qualitative results of multi-part assembly on BreakingBad dataset.

impact of adding the PMT layer to the fine matchers (Tab. 4 (b)). As evident from both tables, incorporating the PMT layer as a fine matcher consistently leads to superior performance, affirming its superiority and importance in this role.

### 4.3 MULTI-PART ASSEMBLY

Multi-part assembly presents a significantly more challenging task, involving establishing multiple matches between fragments and considering the relationships among all these pairwise connections.

We initiate this process by constructing a pose graph for an object with $N$ fractures. This graph comprises relative transformations $T_{ij} = \{R_{ij}|t_{ij}\}$, which act as factors, and individual part fractures $P_i$ as nodes. These relative transformations are computed using the pairwise matcher we have proposed in Sec. 3.1 for pairwise shape assembly. Subsequently, we optimize the pose graph to estimate synchronized global rotations $\tilde{R}_i$ and translations $\tilde{t}_i$. To optimize our pose graph, we employ a state-of-the-art transformation averaging method (Dellaert et al., 2020).

We evaluate the performance of multi-part assembly with the same metrics we used for pairwise shape assembly, but also report the Part Accuracy (PA) (Huang et al., 2020), which is defined as the percentage of fractures with Chamfer Distance less than the predefined threshold, set at 0.01 for our experiments. We present the results of multi-part assembly on the BreakingBad dataset in Tab. 5 and Fig. 4. Notably, our method significantly outperforms all baseline methods. In-depth information on pose graph optimization (PGO) and other implementation specifics are presented in Appendix A.6.

## 5 CONCLUSION

We've developed a new low-complexity, high-order feature transform layer, Proxy Match Transform, designed for efficient approximation of previously compute-intensive layers. This significantly advances the analysis of complex geometric feature correlations while reducing computational load. Despite its excellent performance in geometric shape assembly, the Proxy Match Transform currently focuses on pairwise assembly, indicating potential for future expansion into an efficient method for multi-part assembly.

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

# A APPENDIX

## A.1 THEORETICAL ANALYSIS OF PROXY MATCH TRANSFORM

We now derive sufficient conditions such that Proxy Match Transform can express high-dimensional convolution. Our main theoretical result is given below.

**Theorem 1.** *If we assume* $\mathbf{P}^{(i)\top}\mathbf{P}^{(j)} = \mathbf{I}_{D_{emb}}$ *if* $i = j$ *and* $\mathbf{P}^{(i)\top}\mathbf{P}^{(j)} = \mathbf{0}$ *otherwise for all* $i, j \in [N_h]$, *and define* $\mathbf{A}^{(h)}_{(\mathbf{x},\mathbf{y}),(\mathbf{n},\mathbf{m})} \coloneqq \mathbf{A}^{(h)}_{\mathcal{X}\,(\mathbf{x},\mathbf{n})} \cdot \mathbf{A}^{(h)}_{\mathcal{Y}\,(\mathbf{y},\mathbf{m})}$ *and* $w^{(h)} \coloneqq w^{(h)}_{\mathcal{X}} w^{(h)}_{\mathcal{Y}}$, *then, the dot-product of Proxy Match Transform outputs with a sufficient number of heads* $N_h \geq K^2$ *can express high-dimensional convolutional layer with kernel* $\mathbf{K} \in \mathbb{R}^{K^2}$: $PMT(\mathbf{F}_{\mathcal{X}}) \cdot PMT(\mathbf{F}_{\mathcal{Y}})^{\top} = Conv(\mathbf{F}_{\mathcal{X}}, \mathbf{F}_{\mathcal{Y}}; \mathbf{K})$.

*Proof.* We first take the dot-product of Proxy Match Transform outputs and simplify:

$$\text{PMT}(\mathbf{F}_{\mathcal{X}}) \cdot \text{PMT}(\mathbf{F}_{\mathcal{Y}})^{\top} = \left( \sum_{h\in[N_h]} \mathbf{A}^{(h)}_{\mathcal{X}} \mathbf{F}_{\mathcal{X}} \mathbf{P}^{(h)\top} w^{(h)}_{\mathcal{X}} \right) \left( \sum_{h\in[N_h]} \mathbf{A}^{(h)}_{\mathcal{Y}} \mathbf{F}_{\mathcal{Y}} \mathbf{P}^{(h)\top} w^{(h)}_{\mathcal{Y}} \right)^{\top} \quad (10)$$

$$= \sum_{(i,j)\in[N_h]^2} w^{(i)}_{\mathcal{X}} \mathbf{A}^{(i)}_{\mathcal{X}} \mathbf{F}_{\mathcal{X}} \mathbf{P}^{(i)\top} \mathbf{P}^{(j)} \mathbf{F}^{\top}_{\mathcal{Y}} \mathbf{A}^{(j)\top}_{\mathcal{Y}} w^{(j)}_{\mathcal{Y}} \quad (11)$$

$$= \sum_{(i,j)\in[N_h]^2} \delta(i,j) \left( w^{(i)}_{\mathcal{X}} \mathbf{A}^{(i)}_{\mathcal{X}} \mathbf{F}_{\mathcal{X}} \mathbf{F}^{\top}_{\mathcal{Y}} \mathbf{A}^{(j)\top}_{\mathcal{Y}} w^{(j)}_{\mathcal{Y}} \right) \quad (12)$$

$$= \sum_{h\in[N_h]} w^{(h)}_{\mathcal{X}} \mathbf{A}^{(h)}_{\mathcal{X}} \mathbf{F}_{\mathcal{X}} \mathbf{F}^{\top}_{\mathcal{Y}} \mathbf{A}^{(h)\top}_{\mathcal{Y}} w^{(h)}_{\mathcal{Y}}, \quad (13)$$

where $\delta(i,j)$ provides 1 if $i = j$ and 0 otherwise. Using definitions of $\mathbf{A}^{(h)} \in \mathbb{R}^{|\mathcal{X}||\mathcal{Y}|\times|\mathcal{X}||\mathcal{Y}|}$ and $w^{(h)} \in \mathbb{R}$, the output at a specific position $(\mathbf{x}, \mathbf{y}) \in \mathbb{R}^6$ is as follows:

$$(\text{PMT}(\mathbf{F}_{\mathcal{X}}) \cdot \text{PMT}(\mathbf{F}_{\mathcal{Y}})^{\top})_{(\mathbf{x},\mathbf{y})} = \sum_{h\in[N_h]} \mathbf{A}^{(h)}_{\mathcal{X}\,(\mathbf{x},:)} \mathbf{F}_{\mathcal{X}} \mathbf{F}^{\top}_{\mathcal{Y}} \mathbf{A}^{(h)\top}_{\mathcal{Y}\,(:,\mathbf{y})} w^{(h)} \quad (14)$$

$$= \sum_{h\in[N_h]} \sum_{(\mathbf{n},\mathbf{m})\in\mathcal{X}\times\mathcal{Y}} \mathbf{A}^{(h)}_{\mathcal{X}\,(\mathbf{x},\mathbf{n})} \mathbf{F}_{\mathcal{X}\,(\mathbf{n},:)} \mathbf{F}^{\top}_{\mathcal{Y}\,(:,\mathbf{m})} \mathbf{A}^{(h)\top}_{\mathcal{Y}\,(\mathbf{m},\mathbf{y})} w^{(h)} \quad (15)$$

$$= \sum_{h\in[N_h]} \left( \sum_{(\mathbf{n},\mathbf{m})\in\mathcal{X}\times\mathcal{Y}} \mathbf{A}^{(h)}_{\mathcal{X}\,(\mathbf{x},\mathbf{n})} \cdot \mathbf{A}^{(h)}_{\mathcal{Y}\,(\mathbf{y},\mathbf{m})} \right) \mathbf{C}_{(\mathbf{n},\mathbf{m})} w^{(h)} \quad (16)$$

$$= \sum_{h\in[N_h]} \mathbf{A}^{(h)}_{((\mathbf{x},\mathbf{y}),:)} \mathbf{C}\, w^{(h)}. \quad (17)$$

Now consider the following Lemma:

**Lemma 1.** *Consider a bijective mapping of natural numbers, i.e., heads, onto 6-dimensional local displacements:* $t(h) : [N_h] \to \Delta(\mathbf{x}, \mathbf{y})$. *Let* $\mathbf{A}^{(h)} \in \mathbb{R}^{|\mathcal{X}||\mathcal{Y}|\times|\mathcal{X}||\mathcal{Y}|}$ *be an attention matrix that holds the following:*

$$\mathbf{A}^{(h)}_{(\mathbf{x},\mathbf{y}),(\mathbf{n},\mathbf{m})} = \begin{cases} 1, & \text{if } t(h) = (\mathbf{n}, \mathbf{m}) - (\mathbf{x}, \mathbf{y}) \\ 0, & \text{otherwise.} \end{cases} \quad (18)$$

*Then, for any high-dimensional convolution with a kernel* $K : \mathbb{R}^6 \to \mathbb{R}$, *there exists* $\{w^{(h)} \in \mathbb{R}\}_{h\in[N_h]}$ *such that following equality holds:*

$$\text{Conv}(\mathbf{F}_{\mathcal{X}}, \mathbf{F}_{\mathcal{Y}}; \mathbf{K})_{(\mathbf{x},\mathbf{y})} = \sum_{h\in[N_h]} \mathbf{A}^{(h)}_{((\mathbf{x},\mathbf{y}),:)} \mathbf{C}\, w^{(h)}. \quad (19)$$

*Proof.* Consider high-dimensional convolution at position $(\mathbf{x}, \mathbf{y})$:

$$
\begin{aligned}
\mathrm{Conv}(\mathbf{F}_{\mathcal{X}}, \mathbf{F}_{\mathcal{Y}}; \mathbf{K})_{(\mathbf{x},\mathbf{y})} &:= \sum_{(\mathbf{n},\mathbf{m}) \in \mathcal{N}(\mathbf{x}) \times \mathcal{N}(\mathbf{y})} \mathbf{C}_{(\mathbf{n},\mathbf{m})} K([\mathbf{n} - \mathbf{x}, \mathbf{m} - \mathbf{y}]) \\
&= \sum_{(\boldsymbol{\nu}, \boldsymbol{\mu}) \in \Delta(\mathbf{x},\mathbf{y})} \mathbf{C}_{(\mathbf{x},\mathbf{y}) + (\boldsymbol{\nu}, \boldsymbol{\mu})} K((\boldsymbol{\nu}, \boldsymbol{\mu})) \\
&= \sum_{h \in [N_h]} \mathbf{C}_{(\mathbf{x},\mathbf{y}) + t(h)} K(t(h)) \qquad\qquad ( \; t(h) : [N_h] \to \Delta(\mathbf{x},\mathbf{y}) \; ) \\
&= \sum_{h \in [N_h]} \mathbf{C}_{(\mathbf{x},\mathbf{y}) + t(h)} w^{(h)} \qquad\qquad ( \; w^{(h)} := K(t(h)) \in \mathbb{R} \; ) \\
&= \sum_{h \in [N_h]} \left( \sum_{(\mathbf{n},\mathbf{m}) \in \mathcal{X} \times \mathcal{Y}} \mathbb{1}[t(h) = (\mathbf{n},\mathbf{m}) - (\mathbf{x},\mathbf{y})] \, \mathbf{C}_{(\mathbf{n},\mathbf{m})} \right) w^{(h)} \\
&= \sum_{h \in [N_h]} \mathbf{A}^{(h)}_{((\mathbf{x},\mathbf{y}),:)} \mathbf{C} \, w^{(h)}. \qquad\qquad (20)
\end{aligned}
$$

By applying Lemma 1, we conclude that the dot-product of Proxy Match Transform outputs is equivalent to the high-order convolution. ∎

## A.2 EMPIRICAL ANALYSIS OF PROXY MATCH TRANSFORM

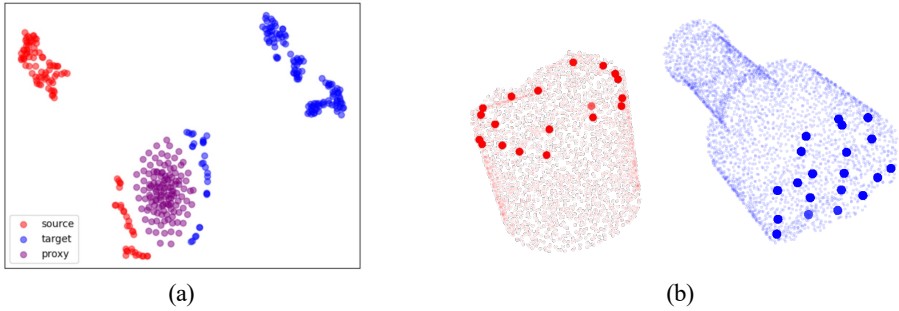

(a)            (b)

Figure 5: Visualization of the correlation between the proxy $\mathbf{P}$ and the input features $\mathbf{F}_{\mathcal{X}}$ and $\mathbf{F}_{\mathcal{Y}}$. The red, blue, and purple color indicates the source ($\mathcal{X}$), target ($\mathcal{Y}$), and the proxy ($\mathbf{P}$), respectively. (a) t-SNE visualization of source feature $\mathbf{F}_{\mathcal{X}}$, target feature $\mathbf{F}_{\mathcal{Y}}$, and shared proxy tensor $\mathbf{P}$. (b) Visualization of points in each point cloud with the highest correlation with the proxy.

To provide the analysis on the role of the shared proxy in our Proxy Match Transform, in Fig. 5, we visualize the correlation between the shard proxy $\mathbf{P}$ and the input features, $\mathbf{F}_{\mathcal{X}}$ and $\mathbf{F}_{\mathcal{Y}}$. As depicted in Fig. 5. (a), the features of the source, target point clouds, and the proxy form distinct clusters, with some regions of the source and target point clouds showing higher correlation with the proxy, resulting in proximity in t-SNE visualization. In Fig. 5. (b), we visually represent those points with high correlation in 3D point clouds. Remarkably, the points with the highest correlation with the proxy are predominantly located on the mating surfaces of the fragments. This observation suggests that the proxy in our Proxy Match Transform effectively facilitates the critical information exchange from the points near the mating surfaces. Note that the correlations are computed on proxy tensor $\mathbf{P}^{(h)}$, specifically with a head index of $h = 0$.

## A.3 Efficiency of Proxy Match Transform

In Tab. 6, we present detailed experimental results that showcase the efficiency of PMT when employed as both a coarse-matcher and a fine-matcher.

Specifically, we measure the computational efficiency of our method by employing Floating Point Operations Per Second (FLOPS) as a metric and compare it with Qin et al. (2022). To assess the memory overhead and footprint, we record the peak memory usage for each method during both the training and inference phases, as well as the number of parameters. For clarity in our comparison, when measuring the FLOPS and the number of parameters, we exclude those associated with the backbone and focus solely on the coarse matcher, and if applicable, the fine matcher.

Table 6: Comparison of computational efficiency and memory usage between Qin et al. (2022) and Proxy Match Transform (PMT).

| Method | Coarse | Fine | FLOPS (G)↓ | # Param. (K)↓ | Mem. train (GB) | Mem. test (GB) |
|---|---|---|---|---|---|---|
| GeoTransformer | GeoTr | - | 9.67 | 926.85 | 6.96 | 3.10 |
| PMT(Coarse) | PMT | - | **0.45** | **273.85** | **2.12** | **0.28** |
| **PMT(Ours)** | PMT | PMT | 0.78 | 296.15 | 3.78 | 0.88 |

The table above clearly indicates that our PMT Blocks deliver substantial reductions not only in computational complexity but also in memory requirements, both during the training and testing phases. This improvement is noteworthy when compared to the previous state-of-the-art methods, specifically Qin et al. (2022). Such efficiency is crucial, as it facilitates the practical deployment of our fine matcher for intricate matching tasks.

## A.4 Implementation Details

**Attention Calculation.** We adopt the relative-position encoding strategy of PerViT (Min et al., 2022) to compute the attention matrix $\mathbf{A}_{\mathcal{X}}^{(h)}$. Specifically, given a query and key positions $\mathbf{q}, \mathbf{k} \in \mathbb{R}^3$, the Euclidean distances is defined as follows: $\mathbf{R}_{\mathbf{q},\mathbf{k}} = ||\mathbf{q} - \mathbf{k}||_2$. An MLP takes this distance to produce an attention score:

$$(\mathbf{A}_{\mathcal{X}}^{(h)})_{\mathbf{q},\mathbf{k}} := \mathrm{Linear}(\mathrm{ReLU}(\mathrm{Linear}(\mathbf{R}_{\mathbf{q},\mathbf{k}}; \mathbf{W}_{\mathrm{p1}}))\mathbf{W}_{\mathrm{p2}}^{(h)})$$
$$= \mathrm{ReLU}(\mathbf{R}\mathbf{W}_{\mathrm{p1}})\mathbf{W}_{\mathrm{p2}}^{(h)} \in \mathbb{R}, \tag{21}$$

where $\mathbf{W}_{\mathrm{p1}} \in \mathbb{R}^{1 \times N_{\mathrm{h}}}$ and $\mathbf{W}_{\mathrm{p2}} \in \mathbb{R}^{N_{\mathrm{h}} \times 1}$ are the linear projection parameters, and ReLU gives non-linearity to the function.

**Training Details.** For the backbone network, we utilize KPConv-FPN (Thomas et al., 2019) and set a subsampling radius of 0.02. And the coarse-level and fine-level feature dimension are set as $D_{\mathrm{c}} = 512$ and $D_{\mathrm{f}} = 128$, respectively. This configuration is employed across all baselines and in our method for both the main results and ablation studies, using a coarse-to-fine approach. For other details, we used the default settings implemented in Qin et al. (2022). We direct readers to their work for further details.

For our PMT blocks with Proxy Match Transform layers, we repeat $N_t = 2$ times to constuct both coarse and fine matchers. In Proxy Match Transform layer, the number of head is set of $N_h = 4$, and the spatial resolutions $D_{\mathrm{proxy}}$ of the proxy tensors are set of 32 for both coarse and fine matcher.

**Baseline methods.** In our ablation studies (Sec. 4), we evaluated our PMT model against four distinct baselines, demonstrating its effectiveness in both coarse- and fine- matching contexts. The first baseline, Linear, involves constructing two individual linear layers for features $\mathbf{F}_{\mathcal{X}}$ and $\mathbf{F}_{\mathcal{Y}}$, sharing a common weight matrix $\mathbf{W} \in \mathbb{R}^{D_{\mathrm{emb}} \times D_{\mathrm{emb}}}$. The second, MLP, employs two linear layers with weight matrices $\mathbf{W}_1 \in \mathbb{R}^{D_{\mathrm{emb}} \times D_{\mathrm{emb}}/2}$ and $\mathbf{W}_2 \in \mathbb{R}^{D_{\mathrm{emb}}/2 \times D_{\mathrm{emb}}}$, each followed by a Group Normalization and ReLU sequence. The third baseline, HDC, adheres to the center-pivot convolution approach as introduced by Min et al. (2021). Lastly, for GeoTr, we implemented the Geometric Transformer according to the method outlined by Qin et al. (2022).

## A.5 EVALUATION METRICS

We employ four different metrics to assess the results. Consider a pair of input point clouds $\{\mathcal{X}, \mathcal{Y}\}$ where $\mathcal{X} \in \mathbb{R}^{N \times 3}$, $\mathcal{Y} \in \mathbb{R}^{M \times 3}$. Without loss of generality, we assume that $N < M$. The ground truth SE(3) relative pose between the point clouds is represented by $\mathbf{T}^{\mathrm{GT}} = \left[\mathbf{R}^{\mathrm{GT}}, \mathbf{t}^{\mathrm{GT}}\right]$, while the predictions are denoted as $\mathbf{T} = [\mathbf{R}, \mathbf{t}]$. Note that in our context, the direction of pose is defined as the transformation that aligns $\mathcal{X}$ with the coordinate frame of $\mathcal{Y}$.

*Chamfer Distance (CD).* The chamfer distance between two point clouds $S_1, S_2$ is defined as

$$d_{\mathrm{CD}}(S_1, S_2) = \sum_{x \in S_1} \min_{y \in S_2} \|x - y\|_2^2 + \sum_{y \in S_2} \min_{x \in S_1} \|x - y\|_2^2, \tag{22}$$

and measures the sum of the distance between nearest neighbor correspondences between point clouds. To assess the quality of shape assembly, we measure the chamfer distance between ground truth assembly and the prediction as:

$$\mathrm{CD}(\mathbf{T}, \mathbf{T}^{\mathrm{GT}}) = d_{\mathrm{CD}}((\mathbf{R}\mathcal{X} + \mathbf{t}) \cup \mathcal{Y}, (\mathbf{R}^{\mathrm{GT}}\mathcal{X} + \mathbf{t}^{\mathrm{GT}}) \cup \mathcal{Y}) \tag{23}$$

*CoRrespondence Distance (CRD).* While the Chamfer distance calculates the distance between two point clouds, its ability to capture more complex features of the object's geometry, such as symmetry and rotation, is limited. To overcome this limitation, we define a new metric, CoRrespondence Distance (CRD). CRD is simply defined as the Frobenius norm between two point clouds (Eq. 24). By considering all pairwise distances between point clouds, it offers a more comprehensive measure of similarity, capturing both proximity and structural alignment.

$$\mathrm{CRD}(\mathbf{T}, \mathbf{T}^{\mathrm{GT}}) = \|(\mathbf{R}\mathcal{X} + \mathbf{t}) \cup \mathcal{Y} - (\mathbf{R}^{\mathrm{GT}}\mathcal{X} + \mathbf{t}^{\mathrm{GT}}) \cup \mathcal{Y}\|_F, \tag{24}$$

*Rotational-, Translational-RMSE (RRMSE, TRMSE).* Finally, to directly measure the prediction accuracy of transformation parameters, we compute the root mean square error (RMSE) between predicted and ground-truth rotation and translation, respectively. Following the protocols of Sellán et al. (2022), we use Euler angle representation for rotation.

## A.6 MULTI-PART ASSEMBLY DETAILS

With ${}^{n}P_2$ computed pairwise transformations, we first construct a pose graph $G = (V, E)$ for an object $O$ with $N$ fractures. We adopt a pose graph optimization (PGO) (Choi et al., 2015) process from the SLAM domain (Carlone et al., 2015b), assuming our estimated relative transformations $R_{ij}|t_{ij}$ to be noisy measurements.

To solve the problem, we divide the pose graph optimization into two steps: (1) rotation averaging, and (2) translation recovering. To estimate global rotation $\tilde{R}_i$ and $\tilde{R}_j$ from estimated relative rotations $R_{ij}$, we set an objective of rotation averaging as follows:

$$\underset{\tilde{R} \in SO(3)}{\arg \min} \sum_{(i,j) \in E} \kappa_{\mathbf{I}_{ij}} \| \tilde{R}_j - \tilde{R}_i R_{ij} \|_F^2, \tag{25}$$

where $\kappa_{\mathbf{I}_{ij}}$ are concentration parameters for an assumed Langevin noise model (Carlone et al., 2015a; Boumal et al., 2014), and $\mathbf{I}_{ij}$ are information matrices. We design our information matrices $\mathbf{I}_{ij}$ as:

$$\mathbf{I}_{ij} = \frac{1}{\| \sum_{(i,j) \in \hat{C}} s_{i,j} \|_2^2} \cdot I_6, \tag{26}$$

with coarse-level correspondence score $s_{i,j} = \exp(- \| \mathbf{h}_i^X - \mathbf{h}_j^Y \|_2^2)$ from each pairwise matcher with input $P_i$ and $P_j$. To optimize our pose graph, we use Shonan Rotation Averaging (Dellaert et al., 2020), the state-of-art transformation optimization method, and we use the implementation in `gtsam` (Dellaert, 2012). Then, we simply recover the global translation $\tilde{t}_i$ and $\tilde{t}_j$ with its objective:

$$\underset{\tilde{t} \in \mathbb{R}^3}{\arg \min} \sum_{(i,j) \in E} \kappa_{\mathbf{I}_{ij}} \| \tilde{R}_i t_{ij} + \tilde{t}_i - \tilde{t}_j \|^2, \tag{27}$$

where $t_{ij}$ are estimated relative translations.

Finally, we assemble the part fractures $P_i$ using transformations derived from synchronized global poses $\{\tilde{R}_i, \tilde{t}_i\}_{i=1}^N$, and we take the system of the largest pieces as canonical pose for the object $O$.

## A.7 ADDITIONAL RESULTS

**Experiments on Real-world Dataset.** In Tab. 7, we have extended our experimental evaluation to include the Fantastic Breaks dataset (Lamb et al., 2023), which consists of real data samples for shape re-assembling. We acknowledge that this dataset is comparatively small, containing only 150 samples, but we believe it provides a valuable preliminary indication of how our model, trained on synthetic data, performs in real-world scenarios.

Table 7: Experimental results on real-world dataset, Fantastic Breaks.

| Method | CRD↓ $(10^{-2})$ | CD↓ $(10^{-3})$ | RMSE (R)↓ (°) | RMSE (T)↓ $(10^{-2})$ | CRD↓ $(10^{-2})$ | CD↓ $(10^{-3})$ | RMSE (R)↓ (°) | RMSE (T)↓ $(10^{-2})$ |
|---|---|---|---|---|---|---|---|---|
| | everyday → Fantastic Breaks | | | | artifact → Fantastic Breaks | | | |
| Global | 26.41 | 16.37 | 88.92 | 22.99 | 26.50 | 17.23 | 87.97 | 24.19 |
| LSTM | 26.48 | 18.53 | 85.26 | 25.00 | 25.85 | 18.25 | 85.18 | 23.29 |
| DGL | 26.92 | 15.22 | 86.66 | 22.76 | 26.23 | 16.98 | 87.96 | 23.58 |
| NSM | 25.05 | 18.62 | 81.88 | 22.54 | 26.09 | 17.28 | 86.69 | 23.36 |
| GeoTransformer | 7.79 | 6.54 | 43.79 | **10.17** | 13.69 | 15.14 | 70.08 | 20.96 |
| **PMT (Ours)** | **7.30** | **6.52** | **39.38** | 11.35 | **12.24** | **13.32** | **66.88** | **19.03** |

**Experiments on Generalizability.** Furthermore, to underscore the generalizability of our approach in Tab. 8, we have conducted transferability experiments between BreakingBad `everyday`, and `artifact` subsets. The results of these experiments confirm that PMT retains its efficacy when applied to data distributions that differ from the training set, indicating robust transfer learning capabilities.

Table 8: Transferability experimental results on BreakingBad dataset.

| Method | CRD↓ $(10^{-2})$ | CD↓ $(10^{-3})$ | RMSE (R)↓ (°) | RMSE (T)↓ $(10^{-2})$ | CRD↓ $(10^{-2})$ | CD↓ $(10^{-3})$ | RMSE (R)↓ (°) | RMSE (T)↓ $(10^{-2})$ |
|---|---|---|---|---|---|---|---|---|
| | everyday → artifact | | | | artifact → everyday | | | |
| Global | 24.81 | 11.82 | 86.55 | 27.50 | 27.34 | 15.24 | 85.67 | 29.09 |
| LSTM | 24.81 | 12.44 | 84.21 | 27.53 | 26.84 | 14.90 | 84.81 | 28.87 |
| DGL | 25.02 | 11.69 | 86.69 | 27.87 | 26.85 | 14.40 | 86.22 | 29.00 |
| NSM | 24.76 | 11.33 | 84.60 | 26.24 | 25.68 | 14.58 | 85.68 | 27.55 |
| GeoTransformer | **3.78** | 3.05 | 61.35 | **15.41** | 4.38 | 3.61 | 61.95 | 14.95 |
| **PMT (Ours)** | 3.96 | **2.97** | **59.02** | 16.31 | **4.20** | **3.33** | **61.25** | 16.06 |

In Tab. 9, we report the classwise quantitative results of pairwise assembly by all methods, evaluated on `everyday` subset of BreakingBad dataset. In Fig. 6 and Fig. 7 we include additional qualitative results of all methods on both `everyday` and `artifact` subset of BreakingBad dataset (Sellán et al., 2022).

Table 9: Class-wise quantitative results of pairwise assembly on `everyday` subset of BreakingBad.

| Method | Beer Bottle | Bowl | Cup | Drinking Utensil | Mug | Plate | Spoon | Teacup | Toy | Wine Bottle | Bottle | Cookie | Drink Bottle | Mirror | Pill Bottle | Ring | Statue | Teapot | Vase | Wine Glass | Sample Mean | Class Mean |
|---|---|---|---|---|---|---|---|---|---|---|---|---|---|---|---|---|---|---|---|---|---|---|
| **CRD ($10^{-2}$) ↓** | | | | | | | | | | | | | | | | | | | | | | |
| ICP | 13.35 | 7.78 | 7.49 | 6.30 | 4.46 | 11.78 | 25.90 | 4.81 | 6.13 | 11.80 | 9.74 | 3.85 | 19.31 | 12.53 | 7.13 | 15.35 | 9.37 | 2.38 | 10.28 | 12.49 | 8.57 | 10.11 |
| SparseICP | 13.33 | 7.73 | 7.27 | 6.17 | 4.42 | 12.00 | 24.90 | 4.90 | 6.16 | 11.69 | 9.75 | 3.91 | 19.25 | 12.69 | 7.09 | 14.95 | 9.25 | 2.36 | 10.18 | 12.74 | 8.55 | 10.04 |
| RANSAC | 14.25 | 9.24 | 6.77 | 5.46 | 4.20 | 13.42 | 21.85 | 5.56 | 5.89 | 8.41 | 8.10 | 4.15 | 17.90 | 12.57 | 7.57 | 16.11 | 8.17 | 3.45 | 10.19 | 11.00 | 8.24 | 9.71 |
| FGR | 11.88 | 8.05 | 6.17 | 5.16 | 4.38 | 11.95 | 25.78 | 5.17 | 6.14 | 8.85 | 8.95 | 4.29 | 17.91 | 13.18 | 6.35 | 15.85 | 8.49 | 2.46 | 9.64 | 12.54 | 8.15 | 9.66 |
| Global | 21.95 | 24.99 | 27.89 | 29.31 | 27.64 | 22.9 | 29.53 | 18.99 | 25.26 | 9.26 | 21.99 | 23.43 | 13.39 | 25.15 | 26.72 | 24.15 | 22.68 | 25.29 | 26.52 | 46.09 | 25.10 | 24.66 |
| LSTM | 30.79 | 24.66 | 27.06 | 28.39 | 28.42 | 23.01 | 27.13 | 17.72 | 27.12 | 23.88 | 27.79 | 22.67 | 30.11 | 24.92 | 28.18 | 24.44 | 22.27 | 24.81 | 31.02 | 27.13 | 27.90 | 26.08 |
| DGL | 16.18 | 24.08 | 28.9 | 29.99 | 28.31 | 23.37 | 27.8 | 18.86 | 26.86 | 8.1 | 16.63 | 22.33 | 8.83 | 25.4 | 26.07 | 19.46 | 19.44 | 22.23 | 25.62 | 53.24 | 24.20 | 23.58 |
| NSM | 25.05 | 22.86 | 21.54 | 25.09 | 25.85 | 22.41 | 26.81 | 14.82 | 24.45 | 9.12 | 17.15 | 21.55 | 13.88 | 24.35 | 25.35 | 22.06 | 17.96 | 26.56 | 23.4 | 18.59 | 1.66 | 21.44 |
| GeoTransformer | 0.62 | 1.84 | 0.86 | 0.22 | 0.70 | 0.97 | 30.35 | 0.32 | 3.23 | 0.56 | 0.83 | 1.06 | 0.77 | 3.73 | 1.47 | 10.79 | 6.16 | 1.01 | 1.41 | 8.24 | 1.66 | 3.76 |
| Ours | 0.51 | 1.51 | 0.48 | 0.78 | 0.76 | 0.89 | 29.41 | 0.45 | 2.74 | 0.28 | 0.76 | 0.35 | 0.47 | 2.48 | 0.66 | 12.37 | 4.14 | 1.14 | 1.64 | 5.36 | 1.46 | 3.36 |
| **CD ($10^{-3}$) ↓** | | | | | | | | | | | | | | | | | | | | | | |
| ICP | 10.88 | 4.66 | 6.11 | 4.65 | 3.38 | 5.15 | 68.63 | 2.10 | 4.07 | 5.78 | 7.16 | 1.26 | 18.58 | 10.25 | 4.13 | 19.56 | 3.90 | 1.81 | 8.53 | 15.65 | 6.52 | 10.31 |
| SparseICP | 10.93 | 5.45 | 6.56 | 4.64 | 3.45 | 9.70 | 69.41 | 2.18 | 4.14 | 5.83 | 7.22 | 1.49 | 18.62 | 15.80 | 4.21 | 19.54 | 3.91 | 1.79 | 8.62 | 15.76 | 7.08 | 10.96 |
| RANSAC | 9.98 | 4.12 | 4.25 | 2.14 | 1.68 | 8.30 | 36.77 | 1.87 | 3.60 | 5.30 | 6.08 | 1.10 | 18.06 | 12.18 | 2.04 | 13.38 | 3.80 | 0.47 | 6.44 | 12.70 | 5.38 | 7.71 |
| FGR | 10.28 | 4.79 | 5.62 | 3.81 | 3.48 | 9.35 | 71.72 | 1.87 | 3.87 | 4.99 | 6.69 | 1.23 | 18.15 | 46.99 | 3.83 | 17.83 | 3.98 | 1.76 | 8.17 | 15.71 | 8.74 | 12.20 |
| Global | 13.87 | 10.39 | 15.42 | 8.56 | 12.0 | 17.11 | 29.64 | 7.45 | 14.11 | 2.2 | 8.48 | 5.35 | 4.75 | 19.1 | 8.53 | 24.56 | 9.56 | 7.04 | 16.39 | 79.69 | 13.80 | 15.71 |
| LSTM | 19.88 | 13.93 | 13.57 | 9.49 | 12.43 | 14.44 | 38.96 | 7.76 | 15.05 | 10.57 | 13.63 | 4.33 | 25.88 | 20.99 | 10.04 | 24.76 | 7.12 | 5.93 | 17.86 | 39.36 | 15.20 | 16.3 |
| DGL | 6.72 | 10.36 | 14.55 | 7.29 | 11.71 | 10.19 | 27.78 | 9.26 | 13.98 | 1.22 | 5.55 | 3.81 | 0.97 | 20.08 | 6.88 | 18.94 | 8.73 | 5.74 | 13.92 | 106.17 | 12.10 | 15.19 |
| NSM | 29.27 | 16.92 | 18.17 | 9.46 | 18.3 | 14.97 | 32.06 | 7.22 | 13.49 | 2.77 | 8.57 | 5.8 | 10.9 | 22.12 | 7.34 | 21.57 | 9.81 | 6.38 | 17.2 | 33.31 | 14.54 | 15.28 |
| GeoTransformer | 0.11 | 0.91 | 0.40 | 0.04 | 0.21 | 0.25 | 69.74 | 0.07 | 2.82 | 0.11 | 0.27 | 0.98 | 0.10 | 4.08 | 0.37 | 4.37 | 4.44 | 0.16 | 0.85 | 13.48 | 1.23 | 5.19 |
| Ours | 0.11 | 0.99 | 0.22 | 0.73 | 0.22 | 0.50 | 72.33 | 0.13 | 2.34 | 0.03 | 0.30 | 0.23 | 0.06 | 2.08 | 0.37 | 15.69 | 4.59 | 0.24 | 1.31 | 4.91 | 1.09 | 5.37 |
| **RMSE (R) (°) ↓** | | | | | | | | | | | | | | | | | | | | | | |
| ICP | 82.15 | 84.10 | 78.21 | 83.58 | 84.51 | 84.12 | 65.96 | 80.86 | 83.64 | 86.82 | 83.45 | 85.22 | 85.35 | 86.79 | 82.60 | 76.50 | 80.76 | 84.35 | 83.64 | 84.53 | 83.71 | 82.36 |
| SparseICP | 82.33 | 83.17 | 76.26 | 83.18 | 84.29 | 88.46 | 47.62 | 81.38 | 83.68 | 85.93 | 84.01 | 86.28 | 85.46 | 85.24 | 80.89 | 76.58 | 79.44 | 84.61 | 83.76 | 83.37 | 83.54 | 81.30 |
| RANSAC | 93.52 | 81.45 | 89.64 | 88.80 | 87.83 | 85.92 | 108.93 | 91.99 | 81.39 | 89.94 | 91.80 | 86.32 | 102.91 | 84.79 | 89.06 | 78.93 | 88.73 | 84.97 | 93.67 | 107.56 | 89.05 | 90.41 |
| FGR | 84.61 | 83.95 | 77.97 | 79.51 | 85.24 | 91.56 | 62.47 | 81.66 | 83.79 | 84.07 | 85.86 | 84.92 | 100.23 | 84.98 | 79.38 | 89.29 | 78.32 | 84.53 | 87.19 | 85.80 | 85.02 | 83.77 |
| Global | 85.5 | 86.81 | 85.56 | 79.97 | 85.13 | 91.28 | 116.8 | 89.09 | 84.23 | 82.89 | 87.04 | 90.95 | 80.18 | 88.21 | 86.89 | 82.14 | 78.7 | 87.15 | 85.97 | 86.52 | 86.40 | 87.05 |
| LSTM | 76.64 | 84.5 | 86.65 | 87.84 | 84.93 | 81.34 | 57.03 | 73.64 | 83.08 | 82.85 | 85.9 | 78.62 | 83.86 | 88.81 | 92.41 | 73.17 | 78.95 | 83.2 | 84.87 | 85.14 | 85.20 | 81.67 |
| DGL | 92.24 | 86.94 | 91.62 | 87.07 | 84.84 | 79.8 | 42.63 | 89.78 | 87.36 | 79.75 | 84.58 | 94.13 | 70.52 | 86.78 | 87.9 | 77.6 | 80.73 | 91.72 | 88.63 | 81.63 | 85.50 | 83.31 |
| NSM | 81.95 | 76.43 | 86.93 | 82.53 | 80.82 | 72.29 | 87.85 | 84.23 | 87.22 | 79.62 | 78.0 | 85.62 | 81.34 | 84.57 | 85.97 | 92.78 | 74.27 | 93.09 | 82.23 | 77.59 | 80.83 | 82.77 |
| GeoTransformer | 7.71 | 25.49 | 26.81 | 6.45 | 40.67 | 22.10 | 98.52 | 3.15 | 77.85 | 16.34 | 27.61 | 24.28 | 4.42 | 34.83 | 25.71 | 69.47 | 97.10 | 68.20 | 21.22 | 64.51 | 34.40 | 38.12 |
| Ours | 7.41 | 23.37 | 16.08 | 14.39 | 49.83 | 18.08 | 115.97 | 12.82 | 63.71 | 11.41 | 26.08 | 11.49 | 2.29 | 29.19 | 31.24 | 54.72 | 42.35 | 76.95 | 22.49 | 64.57 | 32.86 | 34.72 |
| **RMSE (T) ($10^{-2}$) ↓** | | | | | | | | | | | | | | | | | | | | | | |
| ICP | 30.86 | 24.39 | 26.82 | 27.23 | 29.27 | 22.90 | 30.93 | 16.56 | 27.43 | 24.98 | 27.62 | 24.63 | 26.32 | 25.49 | 28.05 | 22.84 | 20.19 | 22.86 | 29.04 | 33.20 | 27.42 | 26.08 |
| SparseICP | 30.77 | 24.15 | 26.17 | 26.63 | 28.87 | 22.70 | 29.99 | 16.79 | 27.43 | 25.17 | 27.64 | 24.70 | 26.21 | 25.63 | 27.83 | 22.28 | 19.94 | 22.73 | 28.83 | 32.94 | 27.26 | 25.87 |
| RANSAC | 31.53 | 26.07 | 23.04 | 20.38 | 26.00 | 19.68 | 27.47 | 18.67 | 26.23 | 21.38 | 25.60 | 21.38 | 27.77 | 27.36 | 22.47 | 18.93 | 20.39 | 33.26 | 28.23 | 36.00 | 26.22 | 25.09 |
| FGR | 27.82 | 24.71 | 25.37 | 25.72 | 28.57 | 22.30 | 30.73 | 16.77 | 26.74 | 22.07 | 27.02 | 24.97 | 25.92 | 26.93 | 26.60 | 25.70 | 18.21 | 23.69 | 28.08 | 33.30 | 26.81 | 25.56 |
| Global | 26.54 | 25.54 | 28.44 | 28.75 | 31.46 | 23.94 | 30.05 | 16.83 | 29.03 | 22.59 | 27.78 | 25.97 | 11.76 | 25.59 | 29.31 | 28.42 | 29.03 | 27.3 | 31.85 | 38.27 | 28.80 | 26.92 |
| LSTM | 36.01 | 25.03 | 30.13 | 30.06 | 32.79 | 22.9 | 38.46 | 17.33 | 30.63 | 29.76 | 33.05 | 24.93 | 25.89 | 26.29 | 32.86 | 25.04 | 23.41 | 26.18 | 33.91 | 41.29 | 30.70 | 29.30 |
| DGL | 22.43 | 25.53 | 29.05 | 27.42 | 28.94 | 23.76 | 31.75 | 16.48 | 29.73 | 19.77 | 25.28 | 24.55 | 8.37 | 26.8 | 29.04 | 22.22 | 22.1 | 26.57 | 30.67 | 33.16 | 28.10 | 25.18 |
| NSM | 30.06 | 24.71 | 25.46 | 29.12 | 28.32 | 22.88 | 34.55 | 19.72 | 29.53 | 14.79 | 18.24 | 23.5 | 11.83 | 26.23 | 28.42 | 23.0 | 17.16 | 21.58 | 25.87 | 16.74 | 24.85 | 23.59 |
| GeoTransformer | 3.74 | 7.14 | 6.52 | 2.31 | 8.23 | 3.07 | 38.69 | 1.41 | 19.56 | 4.68 | 7.60 | 2.87 | 1.48 | 9.61 | 11.65 | 14.58 | 16.48 | 13.01 | 5.37 | 29.88 | 8.82 | 10.39 |
| Ours | 1.37 | 7.47 | 4.92 | 2.54 | 11.05 | 3.46 | 37.60 | 2.35 | 16.20 | 5.03 | 7.63 | 1.87 | 0.92 | 6.53 | 6.66 | 15.73 | 10.82 | 15.81 | 5.52 | 12.52 | 8.17 | 8.80 |

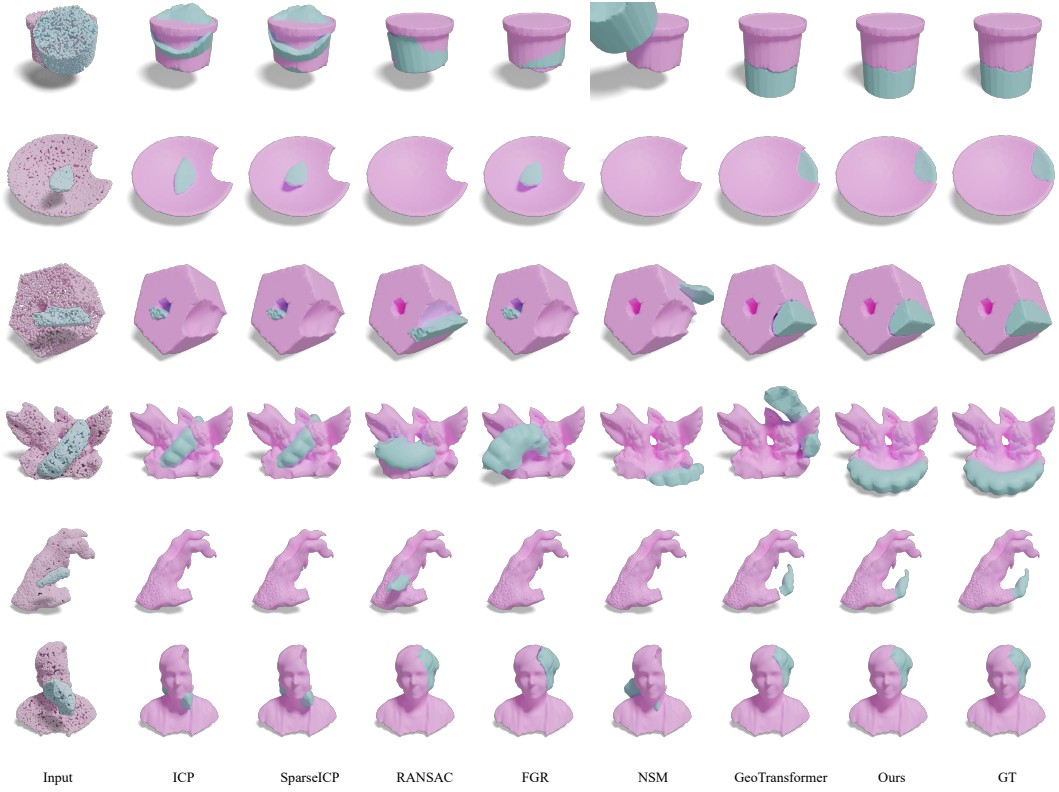

Figure 6: Additional qualitative results of pairwise shape assembly on BreakingBad.

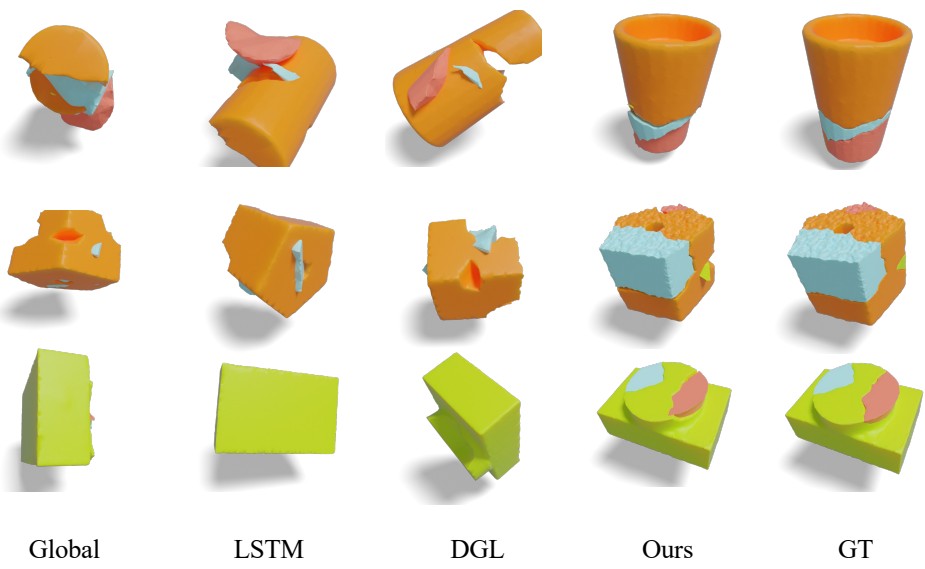

Figure 7: Additional qualitative results of multi-part assembly on BreakingBad.

