# OpenReview forum: "Efficient Point Cloud Matching for 3D Geometric Shape Assembly"
_ICLR.cc/2024/Conference — Submitted to ICLR 2024_

### Official Review · Reviewer_jjsW · 2023-10-30

**Soundness:** 3 good
**Presentation:** 1 poor
**Contribution:** 2 fair
**Rating:** 5
**Confidence:** 4

**Summary:**

This paper introduces Proxy Match Transform (PMT), which is a low-complexity high-order feature transform layer that reduces the computational complexity and memory occupancy. The proposed PMT, combined with a GeoTransformer (CVPR 2022) framework, is used in the task of object assembly, where its effectiveness has been proved by experiments.

**Strengths:**

1. The proposed PMT, at least from the theoretical analysis, could effectively reduce the computational complexity. I personally consider this design similar to KPConv (ICCV 2019), where they use several anchor points to compute their correlations to spatial points that represent the local geometry. This design is reasonable, intuitive, and valuable.

2. The proposed method achieves the state-of-the-art in the benchmark for object assembly, although it is only a synthetic one and there lacks experiments on real data.

3. The Theoretical analysis in the Appendix provides a better understanding of the advantages of the proposed module, which I really appreciate, although it is overly complex to understand throughly.

**Weaknesses:**

1. The whole pipeline is developed upon GeoTransformer (CVPR 2022) and uses a majority of the previous design. The differences are the use of PMT to replace the Self- and Cross-Attention mechanisms in the coare level, as well as the use of PMT after each stage in decoder. I think this weakens the contribution and novelty of this paper.

2. The methodology part is overly complex, and I do not this it is organized well and easy to follow.

    2.1 For example, in Eq. (2) and the following equations, the calculation of attention matrix $\mathbf{A}$ is unclear;

    2.2 Moreover, it is also misleading that the proposed PMT is used to replace the attention mechanisms used in GeoTransformer, while in Fig.1 it is compared to the convolutions. And also in many other places concolutions are introduced, but all the computation of PMT seems like attention-based;

    2.3 In Eq. (2), it seems the output of PMT is the enhaced features, while in Eq. (4), the output is some correlation scores. Do I make a mistake in understanding this?

3. The experiments are only conducted on synthetic dataset, which makes me doubt its value in real applications. Therefore, it is better to include some real data. If there is no real data in this task, as this method is strongly based on GeoTransformer, simply running on GeoTransformer's benchmark also makes sense.

4. As this paper mainly focuses on cutting the memory burden and reducing the complexity. Except for the theoretical analysis, it is also necessary to conduct experiments in terms of the memory occupancy and the running speed, to make comparisons to the state-of-the-art.

**Questions:**

See weaknesses for the questions. I strongly suggest the authors to re-organize their methodology part and simplify their symbols. Fig. 1 does not help understand the main contributions. Also the real-data experiments as well as the comparisons in terms of memory occupancy and running speed are highly encouraged.

---

> ### Author Response · Authors · 2023-11-17
> **Comments to Reviewer jjsW (1/4)**
>
> We appreciate Reviewer jjsW for their insightful comments and suggestions. In response, we have addressed your comments and incorporated revisions into our manuscript, as outlined in the general comments. We encourage the reviewer to check the updated version of our paper.
>
> **[W1] The whole pipeline is developed upon GeoTransformer (CVPR 2022) and uses a majority of the previous design.**
>
> **Answer:** We recognize that the presentation of our method might raise questions about the uniqueness of our technical contribution. It's important to clarify that our approach is built upon the coarse-to-fine matching framework, which serves as a flexible and shared platform for efficient matching across diverse domains [1,2,3,4]. Each method tailors its approach to different matching tasks within this shared platform by designing specific matching layers and model-fitting algorithms. In this context, we underscore our contributions as follows:
>
> Firstly, we introduce Proxy Match Transform, a low-complexity matching layer that effectively refines the matching of the feature pair. Combined with the prevalent coarse-to-fine matching framework, our method outperforms the state-of-the-art methods on the popular object assembly benchmarks while exhibiting order-of-magnitude smaller computational costs.
>
> Secondly, with its substantial reduction in computational cost, the Proxy Match Transform enables its adoption as a fine-level matcher. To the best of our knowledge, this is the first attempt of its kind, refining matching at a granular level. This approach yields state-of-the-art results on recent 3D object assembly benchmarks, where intricate matching is paramount.
>
> Lastly, we theoretically analyze that the Proxy Match Transform layer can effectively approximate the high-order convolutional layers and introduce two sufficient conditions that need to hold.
>
>
> **References.**
> [1] Zhou, Qunjie, Torsten Sattler, and Laura Leal-Taixe. "Patch2pix: Epipolar-guided pixel-level correspondences." Proceedings of the IEEE/CVF conference on computer vision and pattern recognition. 2021.
> [2] Sun, Jiaming, et al. "LoFTR: Detector-free local feature matching with transformers." Proceedings of the IEEE/CVF conference on computer vision and pattern recognition. 2021.
> [3] Yu, Hao, et al. "Cofinet: Reliable coarse-to-fine correspondences for robust pointcloud registration." Advances in Neural Information Processing Systems 34 (2021): 23872-23884.
> [4] Qin, Zheng, et al. "GeoTransformer: Fast and Robust Point Cloud Registration With Geometric Transformer." IEEE Transactions on Pattern Analysis and Machine Intelligence (2023).
>
> ---
>
> **[W2] The methodology part is overly complex, and I do not this it is organized well and easy to follow.**
>
> **Answer:** We acknowledge our method section was not easy to follow, and the concerns raised about its complexity and organization. In response, we have revised Section 3 and added an overview in the early part of the section.
>
> This revision is primarily focused on improving the clarity of notations and presenting the methods in a more structured and unambiguous manner. Additionally, we have revised the section to avoid some omissions in the equations and clarify some confused notations, ensuring that each step is clearly explained and logically follows from the previous one. The revised section also includes additional explanatory text to guide the reader through the methodology, aiming to make it more accessible and easy to follow.
>
> Notably, we've adjusted the notations as follows: $\mathbf{P}$ represents the proxy, same as before. The spatial resolution of the proxy is now denoted by $D_{\text{proxy}}$ rather than $P$. Additionally, we've introduced $\mathcal{X}$ as the notation for the source point cloud, and $\mathcal{Y}$ for the target point cloud.
>
> In the final iteration, we will conduct a comprehensive revision, including any additional comments given during the author-reviewer discussion period.

---

> > ### Author Response · Authors · 2023-11-17
> > **Comments to Reviewer jjsW (2/4)**
> >
> > **[W2.1] For example, in Eq. (2) and the following equations, the calculation of attention matrix  is unclear**
> >
> > **Answer:** We would like to thank the reviewer for the careful reading and the constructive feedback.
> > To enhance readability and organizational clarity, we have included the absent descriptions of the attention matrix in this section. Additionally, we have made revisions to the methodology section of the manuscript. Kindly review the updated version of the manuscript.
> >
> > We adopt the relative-position encoding strategy of PerViT [1] to compute the attention matrix $\mathbf{A}\_{\mathcal{X}}^{(h)}$.
> > Specifically, given a query and key positions $\mathbf{q}, \mathbf{k} \in \mathbb{R}^{3}$, the Euclidean distances is defined as follows: $\mathbf{R}\_{\mathbf{q},\mathbf{k}}=||\mathbf{q}-\mathbf{k}||\_{2}$.
> > An MLP takes this distance to produce an attention score:
> >
> > $$(\mathbf{A}\_{\mathcal{X}}^{(h)})\_{\mathbf{q},\mathbf{k}} \coloneqq \text{Linear}(\text{ReLU}(\text{Linear}(\mathbf{R}\_{\mathbf{q}, \mathbf{k}};\mathbf{W}\_{\text{p1}}))\mathbf{W}\_{\text{p2}}^{(h)})$$,
> > where $\mathbf{W}\_{\text{p1}} \in \mathbb{R}^{1\times N\_{\text{h}}}$ and $\mathbf{W}\_{\text{p2}} \in \mathbb{R}^{N\_{\text{h}} \times 1}$ are the linear projection parameters, and ReLU gives non-linearity to the function.
> >
> > **References.**
> >
> > [1] Juhong Min, Yucheng Zhao, Chong Luo, and Minsu Cho. Peripheral vision transformer. Advances in Neural Information Processing Systems (NeurIPS), 2022.
> >
> > ---
> >
> > **[W2.2] Moreover, it is also misleading that the proposed PMT is used to replace the attention mechanisms used in GeoTransformer, while in Fig.1 it is compared to the convolutions. And also in many other places convolutions are introduced, but all the computation of PMT seems like attention-based;**
> >
> > **Answer:** We apologize for any confusion in our explanation. The relation between convolution and attention is described in the Lemma 1 of the appendix.
> > To clarify it in the main paper, we added the related theorem in the preliminary of Section 3, saying that multi-head self-attention (MHSA) can express convolution, as theoretically proven by the work of Cordonnier et al. (2020) [1].
> > The theorem from their study shows that:
> >
> > **Theorem (Cordonnier et al., 2020).** _A multi-head self-attention layer with_ $N_h$ _heads of dimension_ $D_h$_, output dimension_ $D_{out}$ _and a relative positional encoding of dimension_ $D_p \geq 3$ _can express any convolutional layer of kernel size_ $\sqrt{N_{h}} \times \sqrt{N_{h}}$ _and_ $\min(D_{h},D_{out})$ _output channels._
> >
> > **High-order Convolution as MHSA.** Inspired by this finding, we express high-order convolution at position $(\mathbf{x}, \mathbf{y})$ in an **attention-based** form:
> >
> > $$\text{Conv}(\mathbf{F}\_\mathcal{X}, \mathbf{F}\_\mathcal{Y}; \mathbf{K})\_{(\mathbf{x}, \mathbf{y})} \coloneqq \sum\_{(\mathbf{n}, \mathbf{m}) \in \mathcal{N}(\mathbf{x}) \times \mathcal{N}(\mathbf{y})} \mathbf{C}_{(\mathbf{n}, \mathbf{m})} {K}((\mathbf{n}, \mathbf{m}) - (\mathbf{x}, \mathbf{y})) \\\\
> > = \sum\_{h\in[N\_{h}]}\mathbf{A}\_{((\mathbf{x},\mathbf{y}),:)}^{(h)}\mathbf{C} w^{(h)}$$
> >
> > where $\mathbf{A}$ is an attention matrix, $\mathbf{C}=\mathbf{F}\_{\mathcal{X}}\mathbf{F}\_{\mathcal{Y}}^\top$ is a correlation, $w^{(h)}$ is a learnable weight scalar, and $K: \mathbb{R}^{6} \xrightarrow{} \mathbb{R}$ is a convolutional kernel. This approach is formally supported by the proof provided in Appendix (Sec. A.1, Lemma 1) of our paper, to which we encourage the reviewer to refer for a detailed understanding.
> > In this context, Fig. 1 also illustrates that PMT can significantly reduce computational overhead compared to conventional high-dimensional convolution.
> >
> > **References.**
> > [1] Jean-Baptiste Cordonnier, Andreas Loukas, and Martin Jaggi. On the relationship between self-attention and convolutional layers. In Eighth International Conference on Learning Representations-ICLR 2020, number CONF, 2020.

---

> > > ### Author Response · Authors · 2023-11-17
> > > **Comments to Reviewer jjsW (3/4)**
> > >
> > > **[W2.3] In Eq. (2), it seems the output of PMT is the enhanced features, while in Eq. (4), the output is some correlation scores. Do I make a mistake in understanding this?**
> > >
> > > **Answer:**
> > >
> > > $$\text{PMT}(\mathbf{F}\_{\mathcal{X}}) \coloneqq \sum\_{h \in [{N_h}]} \mathbf{A}\_{\mathcal{X}}^{(h)} \\ \mathbf{F}\_{\mathcal{X}} \\ \mathbf{P}^{(h)\top} {w}\_{\mathcal{X}}^{(h)}.$$
> > >
> > > In Eq. 2 (above), $\text{PMT}(\mathbf{F}_{\mathcal{X}})$ is indeed intended to represent the enhanced feature, rather than any form of correlation score.
> > > The PMT layer computes the output features **based on the correlation** between the input feature and the proxy.
> > >
> > > $$\displaylines{\begin{align} {\text{PMT}(\mathbf{F}\_{\mathcal{X}})^{(h)}}\_{(\mathbf{n}, \mathbf{m})}  &= {\mathbf{A}\_{\mathcal{X}}^{(h)}}\_{(\mathbf{n}, :)}\mathbf{F}\_\mathcal{X} \\ {\mathbf{P}^{(h)\top}} \_{(:,\mathbf{m})}\{w}_{\mathcal{X}}^{(h)} \\\ &= {\mathbf{A}\_{\mathcal{X}}^{(h)}}\_{(\mathbf{n}, :)} {\mathbf{C}\_{\mathcal{X}}^{(h)}}\_{(:, \mathbf{m})} {w}\_{\mathcal{X}}^{(h)} \\\ &= \sum\_{\mathbf{n}' \in \mathcal{N}(\mathbf{n})} {\mathbf{A}\_{\mathcal{X}}^{(h)}}\_{(\mathbf{n}, \mathbf{n}')} {\mathbf{C}\_{\mathcal{X}}^{(h)}}\_{(\mathbf{n}', \mathbf{m})} {w}\_{\mathcal{X}}^{(h)}.\end{align}}$$
> > >
> > > The above equation specifies the output value at head $h$ for a given input feature $\mathbf{F}\_{\mathcal{X}}$ at position $(\mathbf{n},\mathbf{m})$.
> > > It is critical to understand that this does not represent correlation scores, but rather the computed enhanced feature after the application of the attention matrix $\mathbf{A}^{(h)}$ and learnable weight $w^{(h)}$.
> > >
> > > The term $\mathbf{C}_{\mathcal{X}}^{(h)} \coloneqq \mathbf{F}\_{\mathcal{X}} \mathbf{P}^{(h)\top}$ is introduced in this equation as an intermediate computation within each PMT's head. This term does not constitute the final output of PMT; rather, it represents the correlation between the input feature $\mathbf{F}\_{\mathcal{X}}$ and the proxy $\mathbf{P}^{(h)}$.
> > > This intermediate correlation is further processed by PMT's mechanisms to yield the final, enhanced feature representation.

---

> > > > ### Author Response · Authors · 2023-11-17
> > > > **Comments to Reviewer jjsW (4/4)**
> > > >
> > > > **[W3] The experiments are only conducted on synthetic dataset, which makes me doubt its value in real applications.**
> > > >
> > > > **Answer:** We have extended our experimental evaluation to include the Fantastic Breaks dataset (CVPR 2023) [1], which consists of real data samples for shape re-assembling. We acknowledge that this dataset is relatively small, containing only 150 samples. Still, we believe it provides a valuable preliminary indication of our model since it is trained on synthetic data, but it performs in real-world scenarios.
> > > >
> > > > **Table 1: Experimental results on real-world dataset, `everyday` $\rightarrow$ Fanstastic Breaks [1]**
> > > > |Method|CRD $\downarrow$|CD $\downarrow$|RMSE (R) $\downarrow$|RMSE (T) $\downarrow$|
> > > > |:----|:---:|:---:|:---:|:---:|
> > > > ||($10^{-2}$)|($10^{-3}$)|($^{\circ}$)|($10^{-2}$)|
> > > > |Global|26.41|16.37|88.92|22.99|
> > > > |LSTM|26.48|18.53| 85.26|25.00|
> > > > |DGL|26.92|15.22|86.66|22.76|
> > > > |NSM|25.05|18.62|81.88|22.54|
> > > > |GeoTransformer|7.79|6.54|3.79|**10.17**|
> > > > |**PMT (Ours)**|**7.30**|**6.52**|**39.38**|11.35|
> > > >
> > > > **Table 2: Experimental results on real-world dataset, `artifact` $\rightarrow$ Fanstastic Breaks [1]**
> > > > |Method|CRD $\downarrow$|CD $\downarrow$|RMSE (R) $\downarrow$|RMSE (T) $\downarrow$|
> > > > |:----|:---:|:---:|:---:|:---:|
> > > > ||($10^{-2}$)|($10^{-3}$)|($^{\circ}$)|($10^{-2}$)|
> > > > |Global|26.50|17.23| 87.97|24.19|
> > > > |LSTM|25.85|18.25|85.18|23.29|
> > > > |DGL|26.23|16.98|87.96|23.58|
> > > > |NSM|26.09|17.28|86.69|23.36|
> > > > |GeoTransformer|13.69|15.14|70.08|20.96|
> > > > |**PMT (Ours)**|**12.24**|**13.32**|**66.88**|**19.03**|
> > > >
> > > > Furthermore, to emphasize the generalizability of our approach, we have conducted transferability experiments between  BreakingBad `everyday`, and `artifact` subsets. The results of these experiments confirm that PMT retains its efficacy when applied to data distributions that differ from the training set, indicating robust transfer learning capabilities.
> > > >
> > > > **Table 3: Transferability experimental results on BreakingBad `everyday` $\rightarrow$ `artifact`**
> > > >
> > > > |Method|CRD$\downarrow$|CD$\downarrow$|RMSE(R)$\downarrow$|RMSE(T)$\downarrow$|
> > > > |:----|:---:|:---:|:---:|:---:|
> > > > ||($10^{-2}$)|($10^{-3}$)|($^{\circ}$)|($10^{-2}$)|
> > > > |Global|24.81|11.82|86.55|27.50|
> > > > |LSTM|24.81|12.44|84.21|27.53|
> > > > |DGL|25.02|11.69|86.69|27.87|
> > > > |NSM|24.76|11.33|84.60|26.24|
> > > > |GeoTransformer|**3.78**|3.05|61.35|**15.41**|
> > > > |**PMT(Ours)**|3.96|**2.97**|**59.02**|16.31|
> > > >
> > > > **Table 4: Transferability experimental results on BreakingBad `artifact` $\rightarrow$ `everyday`**
> > > >
> > > > |Method|CRD$\downarrow$|CD$\downarrow$|RMSE(R)$\downarrow$|RMSE(T)$\downarrow$|
> > > > |:----|:---:|:---:|:---:|:---:|
> > > > ||($10^{-2}$)|($10^{-3}$)|($^{\circ}$)|($10^{-2}$)|
> > > > |Global|27.34|15.24|85.67|29.09|
> > > > |LSTM|26.84|14.90|84.81|28.87|
> > > > |DGL|26.85|14.40|86.22|29.00|
> > > > |NSM|25.68|14.58|85.68|27.55|
> > > > |GeoTransformer|4.38|3.61|61.95|**14.95**|
> > > > |**PMT (Ours)**|**4.20**|**3.33**|**61.25**|16.06|
> > > >
> > > > **References.**
> > > > [1] Nikolas Lamb, Cameron Palmer, Benjamin Molloy, Sean Banerjee, and Natasha Kholgade Banerjee. Fantastic breaks: A dataset of paired 3d scans of real-world broken objects and their complete counterparts. In Proc. IEEE Conference on Computer Vision
> > > > and Pattern Recognition (CVPR), 2023.
> > > >
> > > > ---
> > > >
> > > > **[W4] it is also necessary to conduct experiments in terms of the memory occupancy and the running speed, to make comparisons to the state-of-the-art.**
> > > >
> > > > **Answer:** We are grateful to the reviewer for highlighting the necessity of presenting empirical evidence to support the efficiency of our proposed method. To address this, we present detailed experimental results that showcase the efficiency of PMT when employed as both a coarse-matcher and a fine-matcher.
> > > >
> > > > Specifically, we measure the computational efficiency of our method by employing Floating Point Operations Per Second (FLOPS) as a metric and compare it with GeoTransformer.
> > > > To assess the memory overhead and footprint, we record the peak memory usage for each method during both the training and inference phases, as well as the number of parameters.
> > > > For clarity in our comparison, when measuring the FLOPS and the number of parameters, we exclude those associated with the backbone and focus solely on the coarse matcher, and if applicable, the fine matcher.
> > > >
> > > > |Method|Coarse matcher|Fine matcher|FLOPS (G)$\downarrow$|\# Param. (K) $\downarrow$|Mem. train (GB)|Mem. test (GB)|
> > > > |-|-|-|-|-|-|-|
> > > > |GeoTransformer|GeoTr|-|9.67|926.85|6.96|3.10|
> > > > |PMT(Coarse)|PMT|-|**0.45**|**273.85**|**2.12**|**0.28**|
> > > > |**PMT(Ours)**|PMT|PMT|0.78|296.15|3.78|0.88|
> > > >
> > > > The table above clearly indicates that our PMT Blocks deliver substantial reductions not only in computational complexity but also in memory requirements, both during the training and testing phases. This improvement is noteworthy when compared to the previous state-of-the-art methods, specifically GeoTransformer. Such efficiency is crucial, as it facilitates the practical deployment of our fine matcher for intricate matching tasks.

---

> > > > > ### Comment · Reviewer_jjsW · 2023-11-22
> > > > > **Comment to the authors**
> > > > >
> > > > > Dear Authors,
> > > > >
> > > > > Thanks again for your detailed response to my questions. I highly appreciate the additional efforts you made expecially in the experiment part to make this manuscript much better and more clear. I think the answers have addressed most of my concerns.
> > > > >
> > > > > However, standing with Reviewer NyfY and jVVT, I also consider the novelty of this paper trivial and probably insufficient for a top-tier conference. It heavily depends on GeoTransformer but tries to optimize memory occupancy using a proxy "trick", which has been widely adopted in previous papers. Also, I have the same feeling as Reviewer jVVT that this paper is hard to follow. Please try to re-organize it and simplify the symbols for the revised version.
> > > > >
> > > > > As a conclusion, I personally would give it a borderline, but it seems there is no such an option. I will consider other reviewrs' opinions and give my final rating.
> > > > >
> > > > > Best,
> > > > > Reviewer jjsW

---

> > > > > > ### Author Response · Authors · 2023-11-23
> > > > > > **Comments to Reviewer jjsW (1/2)**
> > > > > >
> > > > > > We are pleased to hear that most of your concerns are resolved by our rebuttal material. For the remaining issues, we provide further responses for your consideration.
> > > > > >
> > > > > > > **“It heavily depends on GeoTransformer”**
> > > > > >
> > > > > > First of all, we would like to clarify that **the coarse-to-fine matching framework** used in our method is not an original framework of GeoTransformer but it is **widely employed framework in previous matching papers** including recent work of [3,4,6]. While Predator [6], CoFiNet [3], GeoTransformer [4], as well as our PMT, all utilize this common coarse-to-fine framework, each introduces its own coarse matcher, emphasizing distinct contributions. (See table below)
> > > > > >
> > > > > > | Method | Down-sampling | Up-sampling | **Coarse Matcher** | **Fine-grained Matching (Fine Matcher)** | Local correspondence Extractor | Transformation Estimator | Coarse Loss Type | Fine Loss Type |
> > > > > > |:----|:---:|:---:|:---:|:---:|:---:|:---:|:---:|:---:|
> > > > > > | Predator [6] | KPConv | KPConv | **GNN attention** | **No** | Score-based | RANSAC | circle-loss based | Negative Log Likelihood based |
> > > > > > | CoFiNet [3] | KPConv | KPConv  | **GNN attention** | **No** | Optimal Transport [5] | RANSAC & SVD based | leverage weighting scheme | leverage weighting scheme |
> > > > > > | GeoTransformer [4] | KPConv | KPConv  | **Geometric Transformer** | **No** | Optimal Transport [5] | SVD based | circle-loss bassed | Negative Log Likelihood based |
> > > > > > | PMT (Ours) | KPConv | KPConv  | **PMT Block** | **Yes (PMT Block)** | Optimal Transport [5] | SVD based | circle-loss based | Negative Log Likelihood based |
> > > > > >
> > > > > >
> > > > > > For example, Patch2Pix [1] also emphasized the importance of the coarse-to-fine framework in image matching. LoFTR [2] built on this framework and proposed to use transformer layers equipped with self and cross-attention layers to refine coarse correspondences. CoFiNet [3] adapted this idea to a 3D point cloud matching problem with the proposed GNN attention block (as their coarse matcher). GeoTransformer [4] extended this concept by incorporating a geometric transformer (as their coarse matcher), enriching the positional encoding with additional geometric cues.
> > > > > >
> > > > > > Each of these methodologies, **while built on the same coarse-to-fine framework**, has made distinct contributions by **designing specialized matching layers tailored to their specific matching tasks**. In this context of progress, our core contribution is to introduce the Proxy Match Transform (PMT), which is a novel low-complexity feature transform layer that improves geometric matching of two features  by effectively approximating high-order convolution.  Moreover, our method is versatile and can be seamlessly integrated with a wide range of methods other than GeoTransformer. Therefore, **the PMT Block is generic**, so can easily replace many modules for feature matching (like our PMT as coarse matcher), or be added as a module that generates features incorporating correspondence information (like our PMT as fine matcher). In our revision, we will also incorporate results obtained by applying our method to various other techniques (e.g., FCGF, CoFiNet, etc.).
> > > > > >
> > > > > > **References.**
> > > > > > [1] Zhou, Qunjie, Torsten Sattler, and Laura Leal-Taixe. "Patch2pix: Epipolar-guided pixel-level correspondences." Proceedings of the IEEE/CVF conference on computer vision and pattern recognition. 2021.
> > > > > > [2] Sun, Jiaming, et al. "LoFTR: Detector-free local feature matching with transformers." Proceedings of the IEEE/CVF conference on computer vision and pattern recognition. 2021.
> > > > > > [3] Yu, Hao, et al. "Cofinet: Reliable coarse-to-fine correspondences for robust pointcloud registration." Advances in Neural Information Processing Systems 34 (2021): 23872-23884.
> > > > > > [4] Qin, Zheng, et al. "GeoTransformer: Fast and Robust Point Cloud Registration With Geometric Transformer." IEEE Transactions on Pattern Analysis and Machine Intelligence (2023).
> > > > > > [5] Sarlin, Paul-Edouard, et al. "Superglue: Learning feature matching with graph neural networks." Proceedings of the IEEE/CVF conference on computer vision and pattern recognition. 2020.
> > > > > > [6] Huang, Shengyu, et al. "Predator: Registration of 3d point clouds with low overlap." Proceedings of the IEEE/CVF Conference on computer vision and pattern recognition. 2021.

---

> ### Author Response · Authors · 2023-11-21
> **Reminder for Discussion**
>
> Dear reviewer jjsW,
>
> We would like to kindly remind you that the deadline for discussions is approaching. We highly appreciate all the time and effort the reviewers have dedicated to our manuscript, along with their insightful suggestions, and we have thoroughly reviewed and addressed each of the comments you've made. If there are any additional issues, we are ready and willing to continue discussions with the reviewers.
>
> Best regards, Submission 8840 Authors

---

> ### Author Response · Authors · 2023-11-23
> **Comments to Reviewer jjsW (2/2)**
>
> > **“but tries to optimize memory occupancy using a proxy "trick", which has been widely adopted in previous papers.”**
>
> Unfortunately, we are not aware of the "previous papers" you mentioned about. If you specify the papers, we will discuss them compared to ours in revision.
>
> To the best of our knowledge, our approach is unique in the sense that we approximate high-order convolution by introducing a proxy as **a new learnable tensor** that facilitates the exchange of information between two independent feature transforms. It serves as a key computational element within PMT, enabling efficient and effective feature matching in point cloud processing, particularly enhancing the assembly and analysis of 3D geometric shapes. We show both theoretical and empirical results supporting this idea.
>
> In addition, we have enhanced our manuscript with an empirical analysis of proxy tensors and provided detailed descriptions of each element within the PMT layer through our revisions (Appendix A.1, Appendix A.2). We believe these additions will not only help readers better understand how PMT operates but also highlight its novelty by clearly distinguishing how it functions in comparison to other methods using proxies.
>
> ---
>
> > **“Also, I have the same feeling as Reviewer jVVT that this paper is hard to follow. Please try to re-organize it and simplify the symbols for the revised version.”**
>
> We appreciate pointing this out again. In response to the concerns, we have revised the method section in our manuscript to improve its readability. We have also made efforts to simplify the symbols and re-organize the content to make it more accessible to readers. Please refer to our revised manuscript for the changes we have implemented so far, and we will continue to work on further improvements to ensure that the final iteration of our paper is as clear and comprehensible as possible. If you have any specific suggestions for this, we will do our best to reflect them as well.

---

### Official Review · Reviewer_oLtM · 2023-10-31

**Soundness:** 3 good
**Presentation:** 3 good
**Contribution:** 4 excellent
**Rating:** 8
**Confidence:** 2

**Summary:**

High order feature transforms are used to lift features to a high dimensional space to simplify the correlation interpretation. However, this is a computational expensive process. This paper proposes a proxy feature transform(PMT) which transforms the high dimensional feature into an embedding of much smaller dimension while maintaining the feature correlation. As an application, this transform is applied to shape assembly problem in 3D using a coarse-to-fine registration strategy. Experiments show a significant improvement in the performance from existing methods.

**Strengths:**

Strengths:
the paper is well-formalised. It proves the existence of a smaller space orthonormal embedding in the theorem 1, which preserves the convolution of high dimensional features. This can be seen as the PCA for dimension reduction.
The ablation study justifies the various components of the algorithm and the orthonormality condition.
Experimental evaluation seems adequate and clearly indicates the strength of the algorithm.

**Weaknesses:**

No major weakness.

**Questions:**

Few comments
Paper is too compact. More details are required in the proof of theorem 1.
MLP and HDC in table 4 are never mentioned before.
The  transform is learnt on high amount of data. How does it perform with unseen data?

---

> ### Author Response · Authors · 2023-11-17
> **Comments to Reviewer oLtM (1/2)**
>
> We appreciate Reviewer oLtM for their positive comments. We have addressed the comments from reviewers and incorporated revisions into our manuscript, as outlined in the general comments. We encourage the reviewer to check the updated version of our paper. And, these are our answers to some of your questions.
>
> **[Q1] Paper is too compact. More details are required in the proof of Theorem 1.**
>
> **Answer:** We acknowledge our method section was not easy to follow, and the concerns raised about its complexity and organization. In response, we have revised Section 3 and added an overview in the early part of the section.
>
> This revision is primarily focused on improving the clarity of notations and presenting the methods in a more structured and unambiguous manner. Additionally, we have revised the section to avoid some omissions in the equations and clarify some confused notations, ensuring that each step is clearly explained and logically follows from the previous one. The revised section also includes additional explanatory text to guide the reader through the methodology, aiming to make it more accessible and easy to follow.
>
> Notably, we've adjusted the notations as follows: $\mathbf{P}$ represents the proxy, same as before. The spatial resolution of the proxy is now denoted by $D_{\text{proxy}}$ rather than $P$. Additionally, we've introduced $\mathcal{X}$ as the notation for the source point cloud, and $\mathcal{Y}$ for the target point cloud.
>
> In the final iteration, we will conduct a comprehensive revision, including any additional comments given during the author-reviewer discussion period.
>
>
> ---
>
> **[Q2] MLP and HDC in Tab. 4 are never mentioned before.**
>
> **Answer:** In response to the reviewer's query about the MLP and HDC in Table 4, we acknowledge that these baselines were not previously introduced in the main text. We appreciate this opportunity to clarify their roles in our ablation studies, and we updated the revised paper's Appendix to provide a comprehensive explanation of these baselines.
>
> In our ablation studies, we evaluated our PMT model against four distinct baselines, demonstrating its effectiveness in both coarse- and fine-matching contexts. The first baseline, **Linear**, involves constructing two individual linear layers for features $\mathbf{F}\_{\mathcal{X}}$ and $\mathbf{F}\_{\mathcal{Y}}$, sharing a common weight matrix $\mathbf{W} \in \mathbb{R}^{D\_{\text{emb}} \times D\_{\text{emb}}}$. The second, **MLP**, employs two linear layers with weight matrices $\mathbf{W}\_{1} \in \mathbb{R}^{D\_{\text{emb}} \times D\_{\text{emb}}/2}$ and $\mathbf{W}\_{2} \in \mathbb{R}^{D\_{\text{emb}}/2 \times D\_{\text{emb}}}$, each followed by a Group Normalization and ReLU sequence. The third baseline, **HDC**, adheres to the center-pivot convolution approach as introduced by Min et al. (2021) [1]. Lastly, for **GeoTr**, we implemented the Geometric Transformer according to the method outlined by Qin et al. (2022) [2].
>
> **References.**
> [1] Juhong Min, Dahyun Kang, and Minsu Cho. Hypercorrelation squeeze for few-shot segmentation. In Proc. IEEE International Conference on Computer Vision (ICCV), 2021.
> [2] Zheng Qin, Hao Yu, Changjian Wang, Yulan Guo, Yuxing Peng, and Kai Xu. Geometric transformer for fast and robust point cloud registration. In Proc. IEEE Conference on Computer Vision and Pattern Recognition (CVPR), 2022

---

> > ### Author Response · Authors · 2023-11-17
> > **Comments to Reviewer oLtM (2/2)**
> >
> > **[Q3] The transform is learnt on high amount of data. How does it perform with unseen data?**
> >
> > **Answer:** To address concerns about the performance of the Proxy Match Transform (PMT) on unseen data and its applicability in real-world scenarios, we designed two experiments: first, real-world application and second, generalization experiments.
> >
> > First, we have extended our experimental evaluation to include the Fantastic Breaks dataset (CVPR 2023) [1], which consists of real data samples for shape re-assembling.
> >
> > **Table 1: Experimental results on real-world dataset, `everyday` $\rightarrow$ Fanstastic Breaks [1]**
> > |Method|CRD $\downarrow$|CD $\downarrow$|RMSE (R) $\downarrow$|RMSE (T) $\downarrow$|
> > |:----|:---:|:---:|:---:|:---:|
> > ||($10^{-2}$)|($10^{-3}$)|($^{\circ}$)|($10^{-2}$)|
> > |Global|26.41|16.37|88.92|22.99|
> > |LSTM|26.48|18.53| 85.26|25.00|
> > |DGL|26.92|15.22|86.66|22.76|
> > |NSM|25.05|18.62|81.88|22.54|
> > |GeoTransformer|7.79|6.54|3.79|**10.17**|
> > |**PMT (Ours)**|**7.30**|**6.52**|**39.38**|11.35|
> >
> > **Table 2: Experimental results on real-world dataset, `artifact` $\rightarrow$ Fanstastic Breaks [1]**
> > |Method|CRD $\downarrow$|CD $\downarrow$|RMSE (R) $\downarrow$|RMSE (T) $\downarrow$|
> > |:----|:---:|:---:|:---:|:---:|
> > ||($10^{-2}$)|($10^{-3}$)|($^{\circ}$)|($10^{-2}$)|
> > |Global|26.50|17.23| 87.97|24.19|
> > |LSTM|25.85|18.25|85.18|23.29|
> > |DGL|26.23|16.98|87.96|23.58|
> > |NSM|26.09|17.28|86.69|23.36|
> > |GeoTransformer|13.69|15.14|70.08|20.96|
> > |**PMT (Ours)**|**12.24**|**13.32**|**66.88**|**19.03**|
> >
> > Furthermore, to emphasize the generalizability of our approach, we have conducted transferability experiments between  BreakingBad `everyday`, and `artifact` subsets. The results of these experiments confirm that PMT retains its efficacy when applied to data distributions that differ from the training set, indicating robust transfer learning capabilities.
> >
> > **Table 3: Transferability experimental results on BreakingBad `everyday` $\rightarrow$ `artifact`**
> >
> > |Method|CRD$\downarrow$|CD$\downarrow$|RMSE(R)$\downarrow$|RMSE(T)$\downarrow$|
> > |:----|:---:|:---:|:---:|:---:|
> > ||($10^{-2}$)|($10^{-3}$)|($^{\circ}$)|($10^{-2}$)|
> > |Global|24.81|11.82|86.55|27.50|
> > |LSTM|24.81|12.44|84.21|27.53|
> > |DGL|25.02|11.69|86.69|27.87|
> > |NSM|24.76|11.33|84.60|26.24|
> > |GeoTransformer|**3.78**|3.05|61.35|**15.41**|
> > |**PMT(Ours)**|3.96|**2.97**|**59.02**|16.31|
> >
> > **Table 4: Transferability experimental results on BreakingBad `artifact` $\rightarrow$ `everyday`**
> >
> > |Method|CRD$\downarrow$|CD$\downarrow$|RMSE(R)$\downarrow$|RMSE(T)$\downarrow$|
> > |:----|:---:|:---:|:---:|:---:|
> > ||($10^{-2}$)|($10^{-3}$)|($^{\circ}$)|($10^{-2}$)|
> > |Global|27.34|15.24|85.67|29.09|
> > |LSTM|26.84|14.90|84.81|28.87|
> > |DGL|26.85|14.40|86.22|29.00|
> > |NSM|25.68|14.58|85.68|27.55|
> > |GeoTransformer|4.38|3.61|61.95|**14.95**|
> > |**PMT (Ours)**|**4.20**|**3.33**|**61.25**|16.06|
> >
> > **References.**
> > [1] Nikolas Lamb, Cameron Palmer, Benjamin Molloy, Sean Banerjee, and Natasha Kholgade Banerjee. Fantastic breaks: A dataset of paired 3d scans of real-world broken objects and their complete counterparts. In Proc. IEEE Conference on Computer Vision
> > and Pattern Recognition (CVPR), 2023.

---

> ### Author Response · Authors · 2023-11-21
> **Reminder for Discussion**
>
> Dear reviewer oLtM,
>
> We would like to kindly remind you that the deadline for discussions is approaching. We highly appreciate all the time and effort the reviewers have dedicated to our manuscript, along with their insightful suggestions, and we have thoroughly reviewed and addressed each of the comments you've made. If there are any additional issues, we are ready and willing to continue discussions with the reviewers.
>
> Best regards, Submission 8840 Authors

---

> > ### Comment · Reviewer_oLtM · 2023-11-23
> >
> > Dear authors,
> >
> > Thank you for addressing most of my comments in detail. The revised manuscript is indeed more clear in terms of both theoretical formulations and experiments. According to me, the overall message of the paper is that the proposed PMT is interesting due to its computational simplicity and therefore, the ability to perform finer matching.
> >
> > I have also gone through comments from other reviewers. I understand the concerns about the incremental nature of the proposed method. The big question is: what is the significance/impact of the main contribution of the paper, PMT? From what I understand from the paper, the PMT is inspired from the proposed theorem (Cordonnier et al., 2020) relating self-attention to convolution. The authors use this argument to simplify a high-order convolution.
> >
> >  I still think that the PMT has merits of its own to be considered as a novel contribution with significant impact on the performance of coarse-to fine matching (Table 4b) across various methods.  Table 4b also shows that PMT is not as efficient as GeoTransformer (the method it is primarily based on) for coarse matching. However, given its computational simplicity in Table 6, this slight dip in performance seems to be acceptable.
> >
> > Overall, I acknowledge the incremental nature of the proposed method but my opinion about the paper is still fairly positive due to merits of PMT. It is designed to be computationally simpler and the experiments demonstrate convincingly that it mostly outperforms the existing methodologies.

---

> > > ### Author Response · Authors · 2023-11-23
> > > **Reply to reviewer oLtM**
> > >
> > > Dear Reviewer oLtM,
> > >
> > > We would like to express our gratitude for your encouraging feedback and are pleased to know that the issues you raised were effectively addressed in our rebuttal. Your insightful advice has been instrumental in enhancing the clarity of our manuscript, for which we are particularly thankful.
> > >
> > > Furthermore, we appreciate your acknowledgment of the motivation and contributions of PMT. Your understanding aligns with what we have outlined in the first part of our final general comments. We earnestly hope that our responses in the second part of the general comments have also satisfactorily addressed the concerns raised by other reviewers.
> > >
> > > Best regards,
> > > Submission 8840 Authors

---

### Official Review · Reviewer_jVVT · 2023-11-04

**Soundness:** 3 good
**Presentation:** 2 fair
**Contribution:** 3 good
**Rating:** 5
**Confidence:** 4

**Summary:**

This paper proposes a new technique for efficiently aligning surfaces of fractured objects for re-assembly. Its principal insight is that a convolution on a bipartite graph formed from samples on the two object parts (with $N$ and $M$ points respectively) can be simplified (in terms of time/compute resources) by routing the messages that aggregate samples through a proxy transform layer. If the number of heads in this layer is $O(H)$ for some constant $H$, then the presence of the proxy layer changes the quadratic ($N \times M$) complexity of the convolution to $O(H) \times \max{N, M}$ (correct?)

**Strengths:**

The core of the method is a new technique to compute convolutions over the feature sets of the two fragments. The method is theoretically compelling and the authors do a good job of justifying it theoretically and practically. The experimental section seems thorough and validates the method vs several baselines.

**Weaknesses:**

The paper is written in a way that is rather difficult to follow -- I would suggest doing another pass (maybe with feedback from some external readers) to make the language more simple and streamlined.

I am also not entirely sure of the magnitude of the contribution. The proxy match transform is a clever trick that significantly enhances efficiency and accuracy in real-world training scenarios. But it sits inside a large pipeline that draws heavily upon previous work, and it is difficult to gauge its conceptual contribution. Is it a small tweak to make a big system better, or something so critical its impact goes beyond said big system?

For me, the second point puts the paper on the borderline, and the first point pushes it slightly below the threshold. I am open to revising the score based on the rebuttal and other reviewers' comments.

**Questions:**

Exposition:

- Since "shape assembly" is more commonly used to refer to assembling shapes from parts (e.g. a chair from seat, back and legs), it might be clearer to use "shape re-assembly" instead, or even "fractured shape re-assembly".

- "... two sets of features, $\mathcal{F}_P$ and $\mathcal{F}_Q$, associated with each point cloud" --> this reads as: each point cloud has two sets of features. You might want to rephrase as "... two sets of features $\mathcal{F}_P$ and $\mathcal{F}_Q$ associated with the two point clouds respectively" or something like that.

- Please don't use $P$, $\mathbf{P}$ and $\mathcal{P}$ to denote totally different things (near Eq. 3). It's super-confusing.

---

> ### Author Response · Authors · 2023-11-17
> **Comments to Reviewer jVVT (1/2)**
>
> We appreciate Reviewer jVVT for their insightful comments and suggestions. In response, we have addressed your comments and incorporated revisions into our manuscript, as outlined in the general comments. We encourage the reviewer to check the updated version of our paper.
>
> **[W1] The paper is written in a way that is rather difficult to follow.**
>
> **Answer:** We acknowledge our method section was not easy to follow, and the concerns raised about its complexity and organization. In response, we have revised Section 3 and added an overview in the early part of the section.
>
> This revision is primarily focused on improving the clarity of notations and presenting the methods in a more structured and unambiguous manner. Additionally, we have revised the section to avoid some omissions in the equations and clarify some confused notations, ensuring that each step is clearly explained and logically follows from the previous one. The revised section also includes additional explanatory text to guide the reader through the methodology, aiming to make it more accessible and easy to follow.
>
> Notably, we've adjusted the notations as follows: $\mathbf{P}$ represents the proxy, same as before. The spatial resolution of the proxy is now denoted by $D_{\text{proxy}}$ rather than $P$. Additionally, we've introduced $\mathcal{X}$ as the notation for the source point cloud, and $\mathcal{Y}$ for the target point cloud.
>
> In the final iteration, we will conduct a comprehensive revision, including any additional comments given during the author-reviewer discussion period.
>
> ---
>
> **[W2] I am also not entirely sure of the magnitude of the contribution.**
>
> **Answer:** We recognize that the presentation of our method might raise questions about the uniqueness of our technical contribution. It's important to clarify that our approach is built upon the coarse-to-fine matching framework, which serves as a flexible and shared platform for efficient matching across diverse domains [1,2,3,4]. Each method tailors its approach to different matching tasks within this shared platform by designing specific matching layers and model-fitting algorithms. In this context, we underscore our contributions as follows:
>
> Firstly, we introduce Proxy Match Transform, a low-complexity matching layer that effectively refines the matching of the feature pair. Combined with the prevalent coarse-to-fine matching framework, our method outperforms the state-of-the-art methods on the popular object assembly benchmarks while exhibiting order-of-magnitude smaller computational costs.
>
> Secondly, with its substantial reduction in computational cost, the Proxy Match Transform enables its adoption as a fine-level matcher. To the best of our knowledge, this is the first attempt of its kind, refining matching at a granular level. This approach yields state-of-the-art results on recent 3D object assembly benchmarks, where intricate matching is paramount.
>
> Lastly, we theoretically analyze that the Proxy Match Transform layer can effectively approximate the high-order convolutional layers and introduce two sufficient conditions that need to hold.
>
>
> **References.**
> [1] Zhou, Qunjie, Torsten Sattler, and Laura Leal-Taixe. "Patch2pix: Epipolar-guided pixel-level correspondences." Proceedings of the IEEE/CVF conference on computer vision and pattern recognition. 2021.
> [2] Sun, Jiaming, et al. "LoFTR: Detector-free local feature matching with transformers." Proceedings of the IEEE/CVF conference on computer vision and pattern recognition. 2021.
> [3] Yu, Hao, et al. "Cofinet: Reliable coarse-to-fine correspondences for robust pointcloud registration." Advances in Neural Information Processing Systems 34 (2021): 23872-23884.
> [4] Qin, Zheng, et al. "GeoTransformer: Fast and Robust Point Cloud Registration With Geometric Transformer." IEEE Transactions on Pattern Analysis and Machine Intelligence (2023).

---

> > ### Author Response · Authors · 2023-11-17
> > **Comments to Reviewer jVVT (2/2)**
> >
> > **[Q1] Since "shape assembly" is more commonly used to refer to assembling shapes from parts**
> >
> > **Answer:** Thank you for your suggestion to use "shape re-assembly" or "fractured shape re-assembly" in place of "shape assembly." We understand the reasoning behind this suggestion.
> >
> > However, our decision to use the term "geometric shape assembly" was guided by the precedent set in pioneering research by Sell ́an et al. (2022) [1]. In their study, which introduced BreakingBad dataset simulating physically broken objects resulting from external forces, the term "geometric shape assembly" was specifically chosen to describe this area of study.
> >
> > By aligning with the terminology used in Sell ́an et al. (2022) [1], we aim to maintain consistency within the field and ensure clarity for readers who are familiar with this line of research.
> >
> > We hope this explanation clarifies our choice of terminology and its alignment with established research in this area.
> >
> > **References.**
> > [1] Silvia Sell ́an, Yun-Chun Chen, Ziyi Wu, Animesh Garg, and Alec Jacobson. Breaking bad: A dataset for geometric fracture and reassembly. In Thirty-sixth Conference on Neural Information Processing Systems Datasets and Benchmarks Track (NeurIPS), 2022.
> >
> > ---
> >
> > **[Q2] "... two sets of features, $\mathbf{F}\_\mathbf{\mathcal{P}}$ and $\mathbf{F}\_\mathbf{\mathcal{Q}}$, associated with each point cloud" --> this reads as: each point cloud has two sets of features.”**
> >
> > **Answer:** Appreciate the constructive feedback. We've refined the wording to eliminate ambiguity. Kindly consult the updated version of our manuscript for reference.
> >
> > ---
> >
> > **[Q3] Please don't use $P$, $\mathcal{P}$ and $\mathbf{P}$ to denote totally different things (near Eq. 3). It's super-confusing.**
> >
> > **Answer:** Thanks for the suggestions. In response, we've revised the notations within the equations. Notably, we've adjusted the notations as follows: $\mathbf{P}$ represents the proxy, same as before. The spatial resolution of the proxy is now denoted by $D_{proxy}$ rather than $P$. Additionally, we've introduced $\mathcal{X}$ as the notation for the source point cloud (not using $\mathcal{P}$), and $\mathcal{Y}$ for the target point cloud.

---

> ### Author Response · Authors · 2023-11-21
> **Reminder for Discussion**
>
> Dear reviewer jVVT,
>
> We would like to kindly remind you that the deadline for discussions is approaching. We highly appreciate all the time and effort the reviewers have dedicated to our manuscript, along with their insightful suggestions, and we have thoroughly reviewed and addressed each of the comments you've made. If there are any additional issues, we are ready and willing to continue discussions with the reviewers.
>
> Best regards,
> Submission 8840 Authors

---

> > ### Comment · Reviewer_jVVT · 2023-11-23
> > **Borderline**
> >
> > I thank the authors for the careful responses to our questions and for the textual revisions. Unfortunately, they don't fully resolve my concerns. The exposition is still rather unclear -- I still find it very difficult to get real insight into what the method is doing and why it works. And like other reviewers I remain unclear about the magnitude of the contribution. I remain on the borderline -- maybe a little more positive than I was earlier but not fully convinced the contribution is enough -- and clearly explained enough -- for acceptance.

---

> > > ### Author Response · Authors · 2023-11-23
> > > **Comments to Reviewer jVVT (1/2)**
> > >
> > > > **“I still find it very difficult to get real insight into what the method is doing and why it works.”**
> > >
> > > We appreciate your additional questions, and in response to your questions regarding a deeper understanding of our method's inner workings and its effectiveness, we would like to shed more light on the core aspects of our method.
> > >
> > > As highlighted in ***the fourth paragraph of the introduction section*** of our main paper, the core value of our method centers around the introduction of *“a new form of low-complexity high-order feature transform layer, dubbed Proxy Match Transform (PMT)”*. We have dedicated efforts to offer a comprehensive understanding of the role and effectiveness of this proxy, both through theoretical and empirical analysis.
> > >
> > > This begins ***In Section 3 of the main paper and Section A.1 of the appendix*** where we present a theoretical analysis of sufficient conditions such that the Proxy Match Transform can express high-order convolution, and how we enforce these conditions during training via regularization losses, orthonormal loss, and zero loss. To this, we first revisit the concept of high-order convolution, and express this convolution through the multi-head self attention (MHSA), then we show how PMT can effectively express this high-order convolution as MHSA with only sub-quadratic complexity. With this storyline, we include additional explanatory text to guide the reader through the methodology, aiming to make it more accessible and easy to follow.
> > >
> > > ***In Section A.2 of the appendix***, we delve into the empirical analysis of the proxy’s function. This analysis effectively demonstrates that *”the proxy in our Proxy Match Transform effectively facilitates the critical information exchange from the points near the mating surfaces”*, underscoring its importance in our method.
> > >
> > >
> > > To the best of our knowledge, this utilization of the proxy is unique, serving as a key computational element within PMT, enabling efficient and effective feature matching in point cloud processing. In the final iteration, we will conduct a thorough revision to address the points raised by the reviewer and strive to enhance the clarity and comprehensibility of the paper.

---

> > > > ### Author Response · Authors · 2023-11-23
> > > > **Comments to Reviewer jVVT (2/2)**
> > > >
> > > > > **I remain unclear about the magnitude of the contribution**
> > > >
> > > > Again, our central contribution revolves around the introduction of *a new form of low-complexity high-order feature transform layer, dubbed Proxy Match Transform (PMT)”* and the promising experimental results that underscore its effectiveness in shape assembly tasks. At the same time, we would like to clarify that **the coarse-to-fine matching framework** used in our method is not an original framework of GeoTransformer but it is **widely employed framework in previous matching papers** including recent work of [3,4,6]. While Predator [6], CoFiNet [3], GeoTransformer [4], as well as our PMT, all utilize this common coarse-to-fine framework, each introduces its own coarse matcher, emphasizing distinct contributions. (See table below)
> > > >
> > > > | Method | Down-sampling | Up-sampling | **Coarse Matcher** | **Fine-grained Matching (Fine Matcher)** | Local correspondence Extractor | Transformation Estimator | Coarse Loss Type | Fine Loss Type |
> > > > |:----|:---:|:---:|:---:|:---:|:---:|:---:|:---:|:---:|
> > > > | Predator [6] | KPConv | KPConv | **GNN attention** | **No** | Score-based | RANSAC | circle-loss based | Negative Log Likelihood based |
> > > > | CoFiNet [3] | KPConv | KPConv  | **GNN attention** | **No** | Optimal Transport [5] | RANSAC & SVD based | leverage weighting scheme | leverage weighting scheme |
> > > > | GeoTransformer [4] | KPConv | KPConv  | **Geometric Transformer** | **No** | Optimal Transport [5] | SVD based | circle-loss bassed | Negative Log Likelihood based |
> > > > | PMT (Ours) | KPConv | KPConv  | **PMT Block** | **Yes (PMT Block)** | Optimal Transport [5] | SVD based | circle-loss based | Negative Log Likelihood based |
> > > >
> > > >
> > > > For example, Patch2Pix [1] also emphasized the importance of the coarse-to-fine framework in image matching. LoFTR [2] built on this framework and proposed to use transformer layers equipped with self and cross-attention layers to refine coarse correspondences. CoFiNet [3] adapted this idea to a 3D point cloud matching problem with the proposed GNN attention block (as their coarse matcher). GeoTransformer [4] extended this concept by incorporating a geometric transformer (as their coarse matcher), enriching the positional encoding with additional geometric cues.
> > > >
> > > > Each of these methodologies, **while built on the same coarse-to-fine framework**, has made distinct contributions by **designing specialized matching layers tailored to their specific matching tasks**. In this context of progress, our core contribution is to introduce the Proxy Match Transform (PMT), which is a novel low-complexity feature transform layer that improves the geometric matching of two features by effectively approximating high-order convolution.  Moreover, our method is versatile and can be seamlessly integrated with a wide range of methods other than GeoTransformer. Therefore, **the PMT Block is generic**, so can easily replace many modules for feature matching (like our PMT as coarse matcher), or be added as a module that generates features incorporating correspondence information (like our PMT as fine matcher). In our revision, we will also incorporate results obtained by applying our method to various other techniques (e.g., FCGF, CoFiNet, etc.).
> > > >
> > > >
> > > > **References.**
> > > > [1] Zhou, Qunjie, Torsten Sattler, and Laura Leal-Taixe. "Patch2pix: Epipolar-guided pixel-level correspondences." Proceedings of the IEEE/CVF conference on computer vision and pattern recognition. 2021.
> > > > [2] Sun, Jiaming, et al. "LoFTR: Detector-free local feature matching with transformers." Proceedings of the IEEE/CVF conference on computer vision and pattern recognition. 2021.
> > > > [3] Yu, Hao, et al. "Cofinet: Reliable coarse-to-fine correspondences for robust pointcloud registration." Advances in Neural Information Processing Systems 34 (2021): 23872-23884.
> > > > [4] Qin, Zheng, et al. "GeoTransformer: Fast and Robust Point Cloud Registration With Geometric Transformer." IEEE Transactions on Pattern Analysis and Machine Intelligence (2023).
> > > > [5] Sarlin, Paul-Edouard, et al. "Superglue: Learning feature matching with graph neural networks." Proceedings of the IEEE/CVF conference on computer vision and pattern recognition. 2020.
> > > > [6] Huang, Shengyu, et al. "Predator: Registration of 3d point clouds with low overlap." Proceedings of the IEEE/CVF Conference on computer vision and pattern recognition. 2021.

---

### Official Review · Reviewer_NyfY · 2023-11-05

**Soundness:** 3 good
**Presentation:** 3 good
**Contribution:** 3 good
**Rating:** 5
**Confidence:** 2

**Summary:**

The paper proposes Proxy Match transform a low complexity feature transform that can be used in extracting correspondences for shape assembly. This approach tries to solve the quadratic complexity issue of high-order convolution by substituting it with a convolution with a common small-support proxy tensor that captures the local similarities among the shape.

**Strengths:**

Strong experimental results. This paper does not offer a lot of theoretical insight, but if the results are reproducible it will prove useful to a lot of practitioners

**Weaknesses:**

- Lack of motivation: The role and derivation of the both the proxy tensor P and the learnable weight w should be better detailed.
- Limited theoretical insight: the paper feel a bit ad hoc, in the sense of "I have done this and it works." It is not clear how the architecture was derived.

**Questions:**

See points above

---

> ### Author Response · Authors · 2023-11-17
> **Comments to Reviewer NyfY**
>
> We appreciate Reviewer NyfY for their insightful comments and suggestions. In response, we have addressed your comments and incorporated revisions into our manuscript, as outlined in the general comments. We encourage the reviewer to check the updated version of our paper.
>
> **[W1&2]. Lack of motivation & theoretical insight**
>
> **Answer:**
>
> **Motivation and derivation:**
> As outlined in our paper's introduction, high-order feature transform methods have been known to excel in addressing matching problems by capturing structural patterns of correlations in high-dimensional spaces. It is more critical for geometric shape assembly, which requires a meticulous search for pairwise correlations to capture the fine local details of each shape fragment. Despite their efficacy, these methods encounter challenges due to quadratic complexity, limiting their applicability to coarse-grained matching with a restricted number of points. In response to this constraint and with the goal of leveraging a computationally efficient high-order feature transform, we introduce a novel layer termed Proxy Match Transform. We demonstrate its effectiveness and efficiency in the context of shape assembly tasks.
>
> **Role of proxy tensor $\mathbf{P}$:**
> As discussed in Section 3.1 of our paper, the proxy plays a crucial role in achieving the previously mentioned goal of designing an efficient high-order feature transform layer. Essentially, the Proxy Match Transform employs the proxy to reduce the quadratic computational cost of high-order convolution to a sub-quadratic level while effectively expressing the high-order convolution. We have dedicated efforts to analyze the role of the proxy both theoretically and empirically; In Section 3.1, we have discussed the derivation demonstrating how the Proxy Match Transform can express high-order convolution with only sub-quadratic complexity. In Section 3.2, we established sufficient conditions for its approximation. In Section A.1, we have put the extended theoretical analysis of the Proxy Match Transform. In Section A.2, we added the empirical analysis on the proxy and showed how the proxy helps in facilitating critical information exchange between two input point clouds.
>
> **Role of weight $w$:**
> Weight value w is a learnable parameter in Proxy Match Transform layer, serving to accumulate the computation results for each head of the Proxy Match Transform with an appropriate weight.

---

> ### Author Response · Authors · 2023-11-21
> **Reminder for Discussion**
>
> Dear reviewer NyfY,
>
> We would like to kindly remind you that the deadline for discussions is approaching. We highly appreciate all the time and effort the reviewers have dedicated to our manuscript, along with their insightful suggestions, and we have thoroughly reviewed and addressed each of the comments you've made. If there are any additional issues, we are ready and willing to continue discussions with the reviewers.
>
> Best regards,
> Submission 8840 Authors

---

### Author Response · Authors · 2023-11-17
**General Comments**

We thank all the reviewers for their time and valuable insights, as reflected in their thoughtful comments and suggestions. We are pleased to note the positive comments for our work, such as being deemed "theoretically compelling (jVVT)," having "strong experimental results (NyfY)," being "well-formalized (oLtM)," and being considered "reasonable, intuitive, and valuable (jjsW)."

However, we acknowledge the constructive feedback provided by the reviewers, including the following key points:

1. The clarity of the method section's writing needs improvement.
2. The technical novelty of the proposed method could be emphasized further.
3. Additional evaluations on real datasets need to be included.
4. A quantitative analysis of the efficiency of the proposed method is lacking.

In response to this feedback, our rebuttal addresses these concerns by implementing the following revisions:

1. We revised the method section to enhance clarity by improving notations, addressing equation omissions and confused symbols, and adding explanatory text for better accessibility.
2. We added an empirical analysis to better illustrate the role of the proxy
3. We conducted additional experiments, (i) evaluation on real dataset, (ii) transferability experiments to supplement our findings.
4. We included a quantitative analysis of the efficiency of the proposed method to provide a more comprehensive evaluation.

In the final iteration, we are committed to incorporating all feedback from the reviewers, including any additional comments given during the author-reviewer discussion period.

---

### Author Response · Authors · 2023-11-23
**Final General Comments (1/2): Motivation & Contribution**

We are grateful to all reviewers for their dedicated time and valuable feedback, as evidenced by their insightful and constructive comments and suggestions. To further clarify our position, we wish to re-emphasize the motivation and contribution of PMT, aiming to provide reviewers and area chairs with a clearer understanding of our argument.

### I. Motivation of Proxy Match Transform
1) **Why “High-order Convolution”?**
    - **Pros**: With using high-order convolution, we can consider the **6D correlation space**, not the simple feature space.
    - **Limitation**: But despite its good performance, its critical limitation lies in the **quadratic complexity of correlation computation**, i.e., $O(|\mathcal{X}|\cdot|\mathcal{Y}|)$.
    - This offers superior capabilities in terms of capturing complex feature interactions and providing rich contextual information, which is essential for accurate and robust matching.
2) **PMT: Approximate high-order convolution with low complexity**
    - **Purpose**: Take advantage of high-order convolution, but reduce its computational complexity, *i.e.*, quadratic to **sub-quadratic**.
    - **How?**: By incorporating a **shared proxy tensor** in each **independent** feature transform layer of PMT, the matching between feature pairs can be effectively facilitated.
    - **Role of proxy $\mathbf{P}$**: Allow for the exchange of information between features, eliminating the need to construct and convolve memory-intensive pairwise feature correlations, which often contain sparse and limited informative match scores.
3) **Necessity of fine-grained feature matching**
    - **Geometric Assembly**: The task of geometric shape assembly needs verification of **fine local correspondence** between two fragments to sufficiently understand the geometric cues between mating surfaces.
    - Existing point cloud matching methods are unable to address this fine-grained matching due to their inherently **high computational costs**.

---

### II. Contribution of Proxy Match Transform
1) **Approximation of high-order convolution**
    - First approach to approximate the high-order convolution with sub-quadratic complexity.
    - Provide both theoretical and empirical analysis of PMT.
2) **Outstanding efficiency than previous state-of-the-art point cloud matching layer**
    -  PMT is approximately $\times 21.5$ more efficient in FLOPS, $\times 3.38$ compact size on matcher, and needs $\times 3.28$ / $\times 11.07$ less for train / test times compared to GeoTransformer.
|Method|Coarse matcher|Fine matcher|FLOPS (G)$\downarrow$|\# Param. (K) $\downarrow$|Mem. train (GB)|Mem. test (GB)|
|-|-|-|-|-|-|-|
|GeoTransformer|Geometric Transformer|-|9.67|926.85|6.96|3.10|
|**PMT (Coarse)**|PMT Block|-|**0.45**|**273.85**|**2.12**|**0.28**|
|**PMT (Ours)**|PMT Block|PMT Block|0.78|296.15|3.78|0.88|

3) **Importance of fine-grained feature matching**
    - First approach which adopts the fine-grained feature matching in coarse-to-fine point cloud matching platform.
    - Ablation studies (Tab. 4) highlight the necessity of fine-grained matchers and the consequent enhancements in performance they bring.

4) **Generic feature transform layer**
    - Our method is versatile and can be seamlessly integrated with a **wide range of methods**.
    - Can easily **replace many modules** for feature matching (like our PMT as coarse matcher), or **be added as a module** that generates features incorporating correspondence information (like our PMT as fine matcher).

---

> ### Author Response · Authors · 2023-11-23
> **Final General Comments (2/2): Focus on coarse-to-fine point cloud matching framework**
>
> During the author-reviewer discussion period, we have responded to the raised concerns and enhancements to our manuscript. However, while most issues raised have been addressed satisfactorily (as noted by reviewers jjsW, oLtM, and jVVT), there remain a few concerns that seem unresolved, *i.e.*, those dependent on GeoTransformer. Thus, we once again write the response to these issues below and hope this concern can be resolved.
>
> > **“It heavily depends on GeoTransformer”**
>
> > **“And like other reviewers I remain unclear about the magnitude of the contribution.”**
>
> We would like to clarify that **the coarse-to-fine matching framework** used in our method is not an original framework of GeoTransformer but it is **widely employed framework in previous matching papers** including recent work of [3,4,6]. While Predator [6], CoFiNet [3], GeoTransformer [4], as well as our PMT, all utilize this common coarse-to-fine framework, each introduces its own coarse matcher, emphasizing distinct contributions. (See table below)
>
> | Method | Down-sampling | Up-sampling | **Coarse Matcher** | **Fine-grained Matching (Fine Matcher)** | Local correspondence Extractor | Transformation Estimator | Coarse Loss Type | Fine Loss Type |
> |:----|:---:|:---:|:---:|:---:|:---:|:---:|:---:|:---:|
> | Predator [6] | KPConv | KPConv | **GNN attention** | **No** | Score-based | RANSAC | circle-loss based | Negative Log Likelihood based |
> | CoFiNet [3] | KPConv | KPConv  | **GNN attention** | **No** | Optimal Transport [5] | RANSAC & SVD based | leverage weighting scheme | leverage weighting scheme |
> | GeoTransformer [4] | KPConv | KPConv  | **Geometric Transformer** | **No** | Optimal Transport [5] | SVD based | circle-loss bassed | Negative Log Likelihood based |
> | PMT (Ours) | KPConv | KPConv  | **PMT Block** | **Yes (PMT Block)** | Optimal Transport [5] | SVD based | circle-loss based | Negative Log Likelihood based |
>
>
> For example, Patch2Pix [1] also emphasized the importance of the coarse-to-fine framework in image matching. LoFTR [2] built on this framework and proposed to use transformer layers equipped with self and cross-attention layers to refine coarse correspondences. CoFiNet [3] adapted this idea to a 3D point cloud matching problem with the proposed GNN attention block (as their coarse matcher). GeoTransformer [4] extended this concept by incorporating a geometric transformer (as their coarse matcher), enriching the positional encoding with additional geometric cues.
>
> Each of these methodologies, **while built on the same coarse-to-fine framework**, has made distinct contributions by **designing specialized matching layers tailored to their specific matching tasks**. In this context of progress, our core contribution is to introduce the Proxy Match Transform (PMT), which is a novel low-complexity feature transform layer that improves the geometric matching of two features by effectively approximating high-order convolution.  Moreover, our method is versatile and can be seamlessly integrated with a wide range of methods other than GeoTransformer. Therefore, **the PMT Block is generic**, so can easily replace many modules for feature matching (like our PMT as coarse matcher), or be added as a module that generates features incorporating correspondence information (like our PMT as fine matcher). In our revision, we will also incorporate results obtained by applying our method to various other techniques (e.g., FCGF, CoFiNet, etc.).
>
> **References.**
> [1] Zhou, Qunjie, Torsten Sattler, and Laura Leal-Taixe. "Patch2pix: Epipolar-guided pixel-level correspondences." Proceedings of the IEEE/CVF conference on computer vision and pattern recognition. 2021.
> [2] Sun, Jiaming, et al. "LoFTR: Detector-free local feature matching with transformers." Proceedings of the IEEE/CVF conference on computer vision and pattern recognition. 2021.
> [3] Yu, Hao, et al. "Cofinet: Reliable coarse-to-fine correspondences for robust pointcloud registration." Advances in Neural Information Processing Systems 34 (2021): 23872-23884.
> [4] Qin, Zheng, et al. "GeoTransformer: Fast and Robust Point Cloud Registration With Geometric Transformer." IEEE Transactions on Pattern Analysis and Machine Intelligence (2023).
> [5] Sarlin, Paul-Edouard, et al. "Superglue: Learning feature matching with graph neural networks." Proceedings of the IEEE/CVF conference on computer vision and pattern recognition. 2020.
> [6] Huang, Shengyu, et al. "Predator: Registration of 3d point clouds with low overlap." Proceedings of the IEEE/CVF Conference on computer vision and pattern recognition. 2021.

---

### Meta-Review · Area_Chair_zY5o · 2023-12-08

**Metareview:**

This paper examines the task of shape re-assembly i.e. given two point clouds representing pieces of a broken object, predict a relative 6D pose to mend the fracture. As this task requires (implicitly) reasoning about correspondences across the two point clouds, prior works (e.g. GeoTransformer) leverage a transformer to enable this. The key insight of this work is to introduce an efficient ‘proxy match transform’ layer which, instead of an all-pair cross attention, bottlenecks the attention via   some proxy features. The paper also shows some theoretical results where this layer can be shown to approximate higher-order convolutions. The empirical results show the efficacy of this architecture in comparison to the regular transformer baselines.

Despite the empirical results, the paper received mixed ratings with one reviewer recommending acceptance (with low confidence) but the other three leaning towards rejection. In particular, there were several common concerns raised:
a) the technical presentation of the work is rather difficult to follow ( perhaps unnecessarily so), and despite some revisions, would benefit from a rewriting.
b) the magnitude of the contribution is a bit limited as the paper proposes a specific architectural modification in context of an existing pipeline and only examines a specific task. The paper would have been much stronger if the proposed PMS layer could have been shown to be useful in other contexts e.g. point cloud alignment, or and other tasks requiring a higher order convolution.  Moreover, while the analysis relating PMS to a higher-order convolution is appreciated, it would be helpful to understand the empirical benefits over other bottlenecked transformer implementations e.g. Perceiver/PerceiverIO.

Overall, the AC feels that these concerns outweigh the promising results and would lean towards rejection.

**Justification For Why Not Higher Score:**

Overall, the AC feels that the concerns regarding the a) presentation, b) contribution, and c) demonstration for a specific task outweigh the promising results for the studied task.

**Justification For Why Not Lower Score:**

N/A

---

### Decision · Program_Chairs · 2024-01-16

Reject